# Molecular mechanism of plasmid-borne resistance to sulfonamide antibiotics

Meenakshi Venkatesan [1], Michael Fruci[2,3], Lou Ann Verellen[2,3], Tatiana Skarina[1], Nathalie Mesa[1], Robert Flick[1], Chester Pham [1], Radhakrishnan Mahadevan [1,4], Peter J. Stogios [1,5] ✉ & Alexei Savchenko [1,5,6] ✉

The sulfonamides (sulfas) are the oldest class of antibacterial drugs and inhibit the bacterial dihydropteroate synthase (DHPS, encoded by *folP*), through chemical mimicry of its co-substrate *p*-aminobenzoic acid (*p*ABA). Resistance to sulfa drugs is mediated either by mutations in *folP* or acquisition of *sul* genes, which code for sulfa-insensitive, divergent DHPS enzymes. While the molecular basis of resistance through *folP* mutations is well understood, the mechanisms mediating *sul*-based resistance have not been investigated in detail. Here, we determine crystal structures of the most common Sul enzyme types (Sul1, Sul2 and Sul3) in multiple ligand-bound states, revealing a substantial reorganization of their *p*ABA-interaction region relative to the corresponding region of DHPS. We use biochemical and biophysical assays, mutational analysis, and in trans complementation of *E. coli* Δ*folP* to show that a Phe-Gly sequence enables the Sul enzymes to discriminate against sulfas while retaining *p*ABA binding and is necessary for broad resistance to sulfonamides. Experimental evolution of *E. coli* results in a strain harboring a sulfa-resistant DHPS variant that carries a Phe-Gly insertion in its active site, recapitulating this molecular mechanism. We also show that Sul enzymes possess increased active site conformational dynamics relative to DHPS, which could contribute to substrate discrimination. Our results reveal the molecular foundation for Sul-mediated drug resistance and facilitate the potential development of new sulfas less prone to resistance.

Antimicrobial resistance (AMR) is a growing threat to the management of infectious diseases. A recent analysis estimated that more than 1 million deaths were directly caused by and nearly 5 million deaths were associated with antimicrobial-resistant bacterial infections in 2019[1]. AMR is a complex phenomenon involving intrinsically encoded or acquired mechanisms[2]. Antimicrobial resistance-conferring genes (ARGs) often co-localize to mobile genetic elements (MGEs), thereby conferring multidrug resistance and facilitating exchange between species[3]. As the current rate of discovery of new antimicrobial drug classes has been lagging since the golden era of antimicrobial drug discovery in the mid-twentieth century[4–6], new antimicrobial treatment options, including revitalizing old classes of drugs are urgently needed[7–10].

[1]Department of Chemical Engineering and Applied Chemistry, University of Toronto, Toronto, ON M5S 1A4, Canada. [2]London Research and Development Centre, Agriculture and Agri-Food Canada, London, ON N5V 4T3, Canada. [3]Department of Microbiology and Immunology, Western University, London, ON N6A 3K7, Canada. [4]Institute of Biomedical Engineering, University of Toronto, Toronto, ON M5S 3E2, Canada. [5]Center for Structural Biology of Infectious Diseases (CSBID), Calgary, AB, Canada. [6]Department of Microbiology, Immunology and Infectious Diseases, University of Calgary, Calgary, AB T2N 4N1, Canada. ✉e-mail: p.stogios@utoronto.ca; alexei.savchenko@ucalgary.ca

The sulfonamides (sulfas) were the first synthetic compounds deployed as antibacterial drugs. The broad-spectrum bacteriostatic activity of these chemicals against both Gram-positive and Gram-negative bacteria led to their use in both human and veterinary medicine world-wide[11,12]. However, as is the case for other classes of antimicrobials, the rise of resistance has compromised sulfas' clinical utility[13,14]. According to the WHO, 54% and 43% of *Escherichia coli* and *Klebsiella pneumoniae* strains that cause UTIs are highly resistant to co-trimoxazole[15]. With the rise in resistance to front-line treatment regimens such as carbapenems[16], classical antibiotics including sulfas are being re-evaluated for their priority in clinical practice[17,18]. In this respect, a detailed understanding of the molecular basis of sulfa resistance is essential in efforts to extend or revitalize the utility of this drug class.

Sulfas target dihydropteroate synthase (DHPS), encoded by the *folP* gene, an enzyme conserved throughout the bacterial kingdom. DHPS is involved in de novo synthesis of folates, a pathway present only in bacteria and primitive eukaryotes. DHPS catalyzes the condensation of *para*-aminobenzoic acid (*p*ABA) and 6-hydroxymethyl-7,8-dihydropterin pyrophosphate (DHPP), producing 7,8-dihydropteroic acid (7,8-DHP) (Fig. 1a); this compound is further transformed, ultimately leading to tetrahydrofolate, an essential precursor for DNA and RNA synthesis[3]. Sulfas structurally resemble *p*ABA (Fig. 1 and Fig. S1) and thus their mode of action is to directly compete with this co-substrate; the condensation of DHPP and sulfas leads to the formation of a dead-end pterin-sulfa adducts[19–21].

DHPS enzymes adopt the $(\alpha/\beta)_8$ triose phosphate isomerase (TIM) barrel fold which positions the *p*ABA and DHPP-binding sites within the central cavity, with two loops called loops 1 and 2 involved in the catalytic cycle. Detailed analyses of DHPS from *Yersinia pestis* (*Yp*DHPS) and *Bacillus anthracis* (*Ba*DHPS) showed that these loops are involved in pyrophosphate (PP$_i$) binding, coordination of a Mg$^{2+}$ ion and formation of a substructure for binding the *p*ABA molecule[21]. The conformation of the loop 1-loop 2 substructure enabling the binding of *p*ABA requires the presence of pyrophosphate, which was visualized in the crystal structure of the near-transition state complex of *Yp*DHPS[21]. This structural data confirmed that *Yp*DHPS as well as other orthologs operate via a S$_N$1 reaction mechanism, initiated by the binding of DHPP to the active site, elimination of pyrophosphate, leaving a carbocation form of the pterin (DHP$^+$) that reacts with the weakly nucleophilic *p*ABA amine group to form 7,8-DHP[21].

Resistance to sulfas occurs by two mechanisms: mutations in *folP* and/or by acquisition of foreign, sequence-divergent genes coding for DHPS variants that are sulfa-insensitive. The molecular basis of resistance through mutations in *folP* has been investigated in detail and the resistance-conferring substitutions in DHPS have been mapped to loops 1 and 2[21–23]. These substitutions have been demonstrated to increase the $K_M$ parameters for sulfas with less dramatic effects on the $K_M$ for *p*ABA, thus conferring a substrate discrimination ability to these DHPS variants, a property absent in the wild-type (WT) enzymes[22]. The second type of sulfa resistance is associated with genes (*sul*) typically encoded on plasmids found in clinical isolates of such Gram-negative species as *E. coli*, *Acinetobacter baumannii*, and *K. pneumoniae*[24–26]. While this type of plasmid-borne sulfa resistance was first reported in the 1950s and 1960s for *Shigella* and *E. coli*[27,28], it was not characterized until 1975. To date, four mobilizable *sul* genes have been identified: *sul1* was discovered in 1975 in *E. coli* and *Citrobacter* sp.[29,30], *sul2* in 1980 in UTI-causing *E. coli*[31–33], *sul3* in 2003 in *E. coli* from pigs[34,35] and *sul4* in an unknown bacterium present in waste-water in 2017[36]. According

**Fig. 1 | Chemical structures relevant to the reaction catalyzed by the DHPS/Sul enzymes. A** Schematic of folate pathway and chemical structures of sulfonamides and DHPS/Sul ligands. **B** Two-step (S$_N$1) reaction catalyzed by Sul and DHPS enzymes, showing DHP$^+$ intermediate. **C** Structure of 6-HMP (6-hydroxymethylpterin).

**Table 1 | Kinetic parameters for *p*ABA and SMX of Sul and *Ec*DHPS enzymes**

| Enzyme | Native substrate (*p*ABA) | | | Sulfa (SMX) | | | | Fold-change $k_{cat}/K_M$ |
|---|---|---|---|---|---|---|---|---|
| | $K_M$ (µM) | $k_{cat}$ (s$^{-1}$) | $k_{cat}/K_M$ (s$^{-1}$ mM$^{-1}$) | $K_i$ (µM) | $K_M$ (µM) | $k_{cat}$ (s$^{-1}$) | $k_{cat}/K_M$ (s$^{-1}$ mM$^{-1}$) | *p*ABA vs. SMX |
| Sul1$^{WT}$ | 8.41 ± 1.4 | 0.38 | 45.18 | 735 ± 89 | 1004 ± 77 | 0.26 | 0.26 | 173.8 |
| Sul2$^{WT}$ | 9.63 ± 1.2 | 0.46 | 47.77 | 609 ± 140 | 800 ± 58 | 0.28 | 0.35 | 136.5 |
| Sul3$^{WT}$ | 9.66 ± 0.9 | 0.42 | 43.48 | 373.6 ± 77 | 636 ± 67 | 0.25 | 0.39 | 114.5 |
| Sul1$^{F178G}$ | 9.80 ± 1.2 | 0.34 | 34.69 | 3.03 ± 0.3 | 9.4 ± 0.9 | 0.51 | 54.26 | 0.64 |
| Sul3$^{F177G}$ | 9.49 ± 1.1 | 0.47 | 49.53 | 4.16 ± 0.3 | 9.3 ± 0.9 | 0.49 | 52.69 | 0.94 |
| *Ec*DHPS | 7.82 ± 0.6 | 0.38 | 48.59 | 5.15 ± 1.1 | 7.67 ± 0.7 | 0.46 | 59.97 | 0.81 |

*p*ABA and SMX kinetic parameters of Sul and *Ec*DHPS enzymes. The substrate turnover $k_{cat}$ and catalytic efficiency $k_{cat}/K_M$ was calculated from the $V_{max}$ and $K_M$ obtained from the Michaelis-Menten non-linear regression using GraphPad Prism v5.0. Error represents SD for the mean value (mean of biological replicates of *n* = 3) for each *p*ABA/SMX concentration.

to the CARD Database[37], 40% of *Pseudomonas aeruginosa*, 16% *Enterobacter cloaceae*, 18% *K. pneumoniae* and 44% of *A. baumannii* genomes contain *sul1*, reflecting its high degree of spread. *sul* genes are often part of multiple resistance gene clusters[38–43]. Also according to the CARD database, *sul* genes are particularly well disseminated in some environmental species such as *Comamona testosteroni* and *A. towneri*[44,45]. *sul1* is regularly tracked as a surrogate marker for the dissemination of ARGs and anthropogenic influence on the environment, i.e., waste-water treatment and decontamination of ARGs[46–49]. It was recently observed that sulfamethoxazole (SMX) represents one of the largest sources of pollution of the world's rivers[50]; this may exert a strong selective pressure on bacteria to acquire and maintain the *sul* genes.

A recent study suggested that the *sul* genes evolved from lateral transfer of chromosomal *folP* genes of *Rhodobiaceae* and *Leptospiraceae* species[51]. However, the direct ancestor of *sul* or the mechanism of their capture onto MGEs have not been identified. Consistent with the notion that *sul* genes evolved from *folP* within environmental microbiota, two other studies using functional metagenomics sampling of soil or agricultural environments identified divergent *folP* genes that also confer sulfa resistance[52,53].

Despite sharing significant sequence similarity with DHPS (~30% sequence identity with the *E. coli* DHPS enzyme, *Ec*DHPS), the Sul enzymes demonstrate the ability to discriminate between *p*ABA and sulfas as substrates. The first observation of this capability occurred from a comparison of DHPS activity from sulfa-resistant vs sulfa-sensitive *E. coli*[32]. However, there have been no molecular studies of Sul enzymes which would shed light on the mechanism by which sulfa resistance is conferred by these enzymes.

Here, we employ structural analysis, enzymatic and fluorescence assays, antimicrobial susceptibility testing using *E. coli* Δ*folP*, and adaptive evolution of *E. coli* to reveal the molecular basis of sulfa resistance by the plasmid-encoded Sul1, Sul2, and Sul3 enzymes. We show that the Sul enzymes feature a remodeled *p*ABA-binding region and demonstrate different conformational dynamics in their active sites as compared to DHPS. Our data reveals that Sul enzymes possess an additional phenylalanine residue lacking in DHPS that is positioned to block sulfonamide binding, thus representing the key molecular element in Sul-mediated sulfonamide resistance.

## Results

### Despite sequence divergence, Sul enzymes demonstrate catalytic properties like those of DHPS

We first characterized the catalytic properties of Sul enzymes in comparison to DHPS. We recombinantly purified the Sul1, Sul2 and Sul3 enzymes as well as *Ec*DHPS. We characterized the activity of these four enzymes in vitro by measuring the release of PP$_i$. In the presence of excess *p*ABA, saturation of DHPP could not be reached and thus we were not able to obtain the kinetic parameters for DHPP

under these experimental conditions[21,54]. Thus, we proceeded to characterization of the enzymes' kinetic properties with respect to the *p*ABA co-substrate in the presence of excess DHPP. All three Sul enzymes and *Ec*DHPS demonstrated similar $K_M$ values for *p*ABA (Table 1 and Fig. S2). Accordingly, the three Sul and *Ec*DHPS enzymes showed no significant difference in the catalytic efficiency ($k_{cat}/K_M$) for the *p*ABA co-substrate. These results suggested that these Sul enzymes can effectively replace the activity of DHPS in folate biosynthesis in *E. coli*.

### Sulfa drugs fail to inhibit the Sul enzymes

Previous comparative analysis of the Sul and *Ec*DHPS enzymes also suggested similar affinity for these enzymes toward the *p*ABA co-substrate[32]. While performed with only partially purified protein samples, this study concluded that sulfathiazole is 10,000-fold less effective in abrogating the activity of the Sul enzyme compared to *Ec*DHPS[32]. We tested whether SMX as a representative sulfa drug acts as an inhibitor of the purified Sul enzymes, and, whether it is co-metabolized with DHPP in the dihydropteroate synthase reaction forming a dead-end adduct[19–21]. We first confirmed SMX inhibition of the activity of *Ec*DHPS in an in vitro assay against DHPP and *p*ABA (Fig. S3). According to our results, the activity of *Ec*DHPS was inhibited by SMX with a $K_i$ of 5.1 µM. This value is similar to this enzyme's $K_M$ for *p*ABA (Table 1 and Fig. S3), reflecting its inability to discriminate between the substrate and inhibitor. Next, we tested Sul activity against SMX and DHPP but in the absence of *p*ABA. Further reflecting SMX's competitive mechanism of inhibition, we identified the catalytic efficiency of *Ec*DHPS for SMX and *p*ABA as 49 and 60 s$^{-1}$ mM$^{-1}$, respectively (Table 1 and Fig. S3). For *Ec*DHPS, we also validated the formation of the pterin-SMX adduct in both oxidized and reduced forms (Fig. S4) in the presence of at least 4 µM SMX.

The dihydropteroate synthase activities of Sul enzymes are inhibited by SMX with $K_i$ values much lower (between 143 to 73-fold) than in the case of *Ec*DHPS (Table 1). In the assay against DHPP and SMX (no *p*ABA), Sul1 demonstrates 131-fold less affinity ($K_M$) to SMX, while Sul2 and Sul3 showed 104 and 83-fold decrease in affinity, respectively, relative to *Ec*DHPS (Table 1). Similarly, we identified that the catalytic efficiency values ($k_{cat}$) for SMX as the co-substrate versus *p*ABA are much lower for the Sul enzymes relative to *Ec*DHPS (between 111 to 137-fold lower). Finally, we detected the formation of the pterin-SMX adduct catalyzed by Sul1, but its formation was only detected at much higher concentrations of SMX (>500 µM) as compared to *Ec*DHPS (Fig. S4).

Altogether, these results unambiguously confirm the inability of sulfa drugs such as SMX to inhibit Sul enzymes despite their dihydropteroate synthase enzymatic properties being similar to those of DHPS enzymes. The enzymatic analysis suggested that this divergence is based on the ability of the Sul enzymes to discriminate between *p*ABA and SMX compounds.

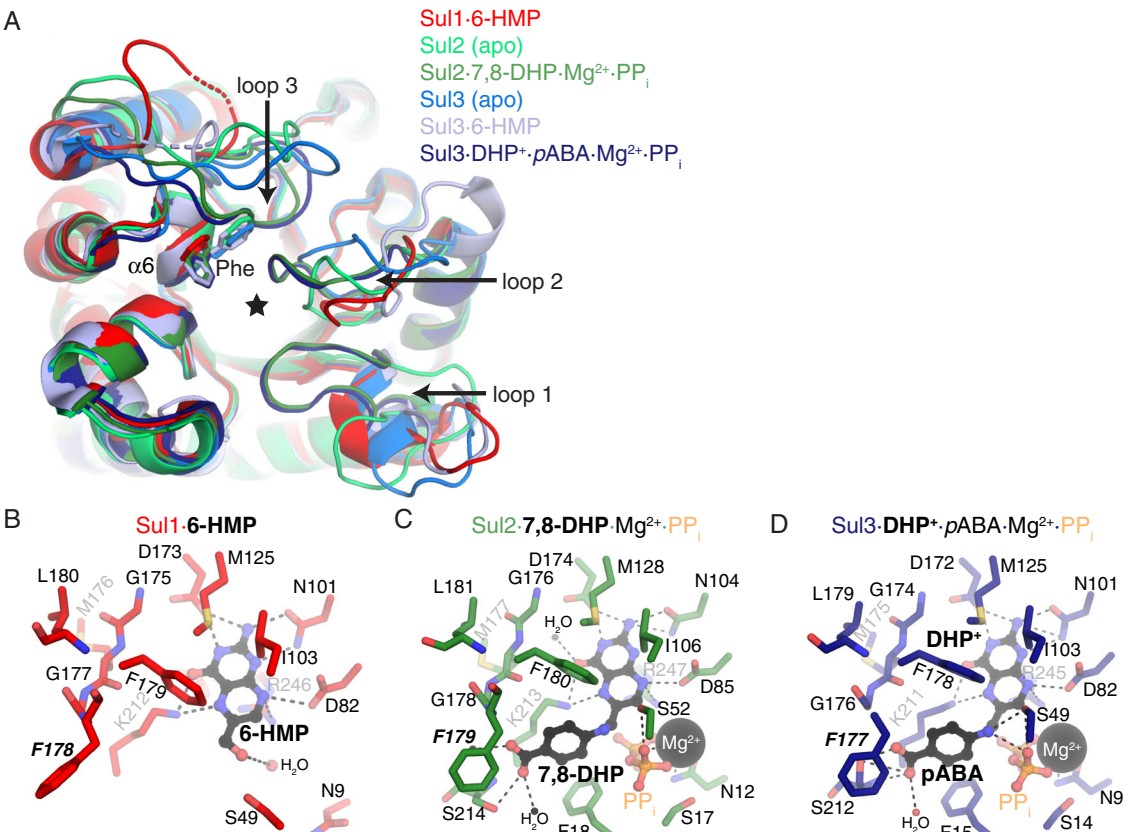

**Fig. 2 | Crystal structures of Sul enzymes in various ligand-bound states.**
**A** Superposition of the Sul enzyme structures determined in this study. Loops 1 through 3 are labeled. The active site Phe that confers sulfonamide resistance is shown in stick representation and labeled. Star indicates location of catalytic site.

α6 region is labeled. **B** Catalytic site of the Sul1 enzyme. **C** Catalytic site of the Sul2 enzyme. **D** Catalytic site of the Sul3 enzyme. For B through D: bound ligands are shown in black ball-and-stick, $Mg^{2+}$ ions in spheres, the active site Phe that confers sulfonamide resistance is labeled in bold italics.

## Crystal structures of Sul enzymes reveal active features responsible for ligand discrimination

We determined the crystal structure of Sul1 in a ligand-bound state, and the Sul2 and Sul3 enzymes in both *apo* and ligand-bound forms to resolutions between 1.74 to 2.8 Å (Table S1).

All three Sul enzymes adopted the $(\alpha/\beta)_8$ TIM barrel fold (Fig. 2a) also shared by DHPS (Fig. 3). The active site cleft is located to the solvent-exposed side of the barrel and penetrates deep into its center. The substrate binding cleft in the Sul structures is capped by three loops: loop 1 connecting α1 and β1 (corresponding to residues 10-23 in Sul1), loop 2 connecting α2 and β2 (residues 46-56 in Sul1), and loop 3 connecting α5 and β5 (residues 127–140 in Sul1) (Fig. S5).

Comparison of Sul *apo* and ligand-bound structures revealed conformational changes in these three loops triggered by ligand binding (Fig. 2a, b). While each crystal adopted different space groups, the loops largely did not participate in crystal contacts or block active sites, except for the Sul1 *apo* structure where loops 1 and 2 interdigitated into the active site of other chains in the crystal lattice (Fig. S6). In particular, loop 1 is oriented further away from the ligand-binding cleft in the *apo* structures of Sul2 and Sul3 as compared to the structures of these enzymes in 7,8-DHP or *p*ABA-bound states (Fig. 2). Similarly, loop 3 in Sul2 and Sul3 also undergoes a large conformational change upon binding of *p*ABA or 7,8-DHP (Fig. 2). Finally, in the Sul2 and Sul3 structures, loop 2 undergoes conformational changes in a more confined space in presence of the ligands (Fig. 2). Based on the active role of the loops 1 and 3 in substrate binding, we hypothesized that these structural elements may be involved in discrimination against sulfas.

We noticed that the Phe residue that is conserved across Sul enzymes (Phe178 in Sul1, Phe179 in Sul2 and Phe177 in Sul3) and

localized to the active site adopted different rotamer conformations depending on ligand binding (Fig. 2). In general, the sidechain of this residue and its environment (α6) showed higher crystallographic *B*-factors than the average protein (i.e., for the Sul2 apo structure, *B*-factor of Phe179 sidechain atoms were 32 to 67, for α6: 20 to 53, and for the protein average: 36). Based on these observations we suggested that this conserved Phe residue may be important for substrate discrimination.

We identified electron density in the active site of Sul1 corresponding only to a pterin ring which we modeled as 6-hydroxymethylpterin (6-HMP, Fig. 1 and Fig. S6a). The 6-HMP molecule formed hydrogen bonds with the Sul1 residues Asn101, Asp173 and Lys212. Additionally, the plane of the pterin ring formed van der Waals interactions with Phe179 and Ile103.

The Sul2·7,8-DHP·$Mg^{2+}$·$PP_i$ complex structure provided essential insights into the molecular architecture of the Sul enzyme's fully ligand-occupied active state, containing both products of the enzymatic reaction and with loops 1, 2 and 3 engaged in ligand interactions (Fig. 2c and Fig. S6). In addition to interactions observed between Sul1 and 6-HMP, the Sul2 complex structure revealed interactions contributed by loops 1 residues Asn12, Ser17, Phe18 and the loop 2 residue Ser52, consistent with the role of these loops in $PP_i$ release. In comparison to the Sul1 structure, the *p*ABA molecule formed fewer interactions with the Sul2 enzyme and occupied the more solvent-exposed positively charged region of the active site pocket. In this conformation, the *p*ABA carboxylate forms interactions with Ser214, the *p*-amino group with Ser52, and with aliphatic region of Lys213 and the sidechain of Phe179.

The Sul3·DHP⁺·$Mg^{2+}$·$p$ABA complex structure also featured a well-defined active site. This Sul complex structure contained electron

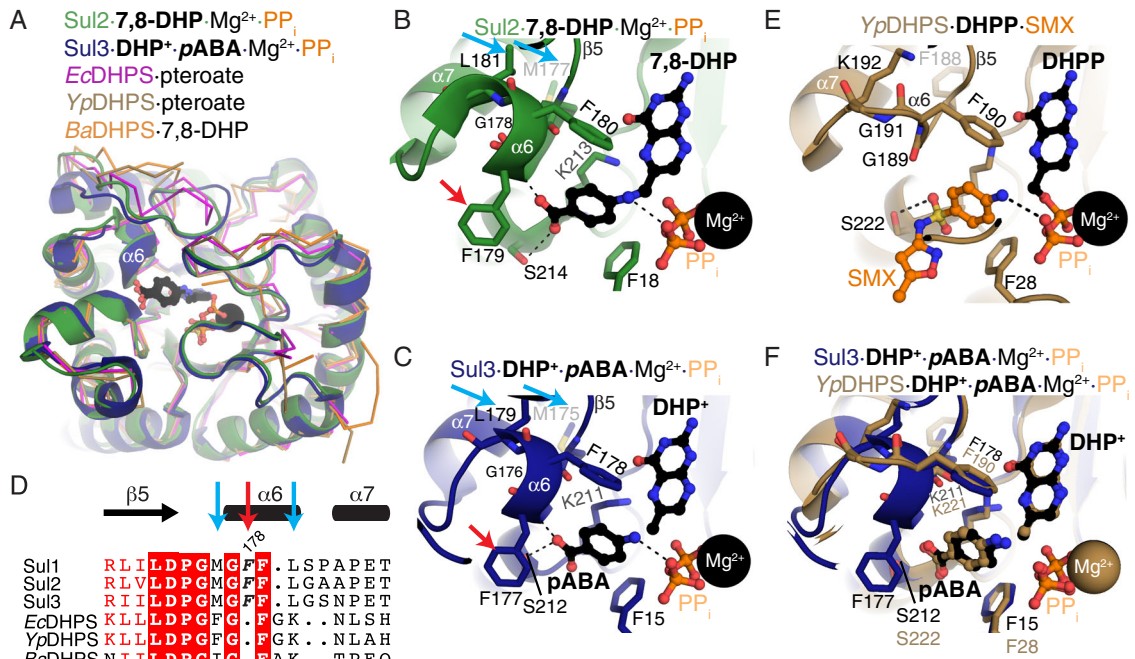

**Fig. 3 | Comparison of the structures of Sul and FolP enzymes. A** Superposition of the Sul2·7,8-DHP·Mg²⁺·PPᵢ complex, Sul3·DHP⁺·pABA·Mg²⁺·PPᵢ complex, *Cb*FolP·pteroate complex (PDB 3tr9), *Ba*FolP·7,8-DHP complex (PDB 3tya), *Yp*FolP·pteroate complex (PDB 3tyu). **B** Zoom into active sites of the Sul2 and **C** Sul3 enzymes. **D** Structure-based multiple sequence alignment of Sul and FolP enzymes. White text/red background represents full sequence identity, red text/white background indicates high sequence similarity. Red arrow and bold+italics representation indicates Phe insertion in Sul enzymes and blue arrow indicates substitutions to hydrophobic residues nearby the Phe insertion (Met and Leu) in Sul enzymes. Number 178 refers to Sul1 numbering. Secondary structure elements are shown above the alignment. **E** Zoom into the active site of *Yp*FolP·DHPP·SMX complex, PDB 3tzf. For B and C, amino acids shown in sticks are those that interact with *p*ABA and/or SMX, and ligands are shown in ball-and-stick and Mg²⁺ ions are shown as black spheres. α6 and β5 regions are labeled. **F** Overlay of Sul3·DHP⁺·pABA·Mg²⁺·PPᵢ complex and *Yp*FolP·DHP⁺·pABA·Mg²⁺·PPᵢ complex (PDB 3tyz).

density corresponding to DHP⁺²¹, *p*ABA, Mg²⁺ and PPᵢ (Fig. S6d). Multiple electron density features verify this trapped molecule is DHP⁺, including a clear break between its C9 position and the *p*-amino group of *p*ABA, and a ring pucker around atoms C6, C7 and N8 consistent with saturation at the C7 position (Fig. S6d). These observations are consistent with *in crystallo*-capture of the intermediate of the S$_N$1 enzymatic reaction mechanism, wherein the PPᵢ group has been cleaved from DHPP but the *p*ABA molecule had not yet been ligated to DHP⁺²¹.

The position of DHP⁺ in the Sul3·DHP⁺·pABA·Mg²⁺·PPᵢ complex superimposed with its position in the *Yp*DHPS transition state complex structure (Fig. 3f)²¹ and the pterin ring in the Sul2·7,8-DHP·Mg²⁺·PPᵢ complex. All the protein-ligand interactions observed in the Sul2·7,8-DHP·Mg²⁺·PPᵢ complex were consistent with those observed in the Sul3·DHP⁺·pABA·Mg²⁺·PPᵢ complex, including the hydrophobic interaction between *p*ABA and Phe177.

**Interaction with pABA triggers remodeling in Sul enzyme's active site involving the key phenylalanine residue**

A comparison of the structures of DHPS from *Yersinia pestis* and *Bacillus anthracis*²¹,⁵⁵ with the Sul2·7,8-DHP·Mg²⁺·PPᵢ or the Sul3·DHP⁺·pABA·Mg²⁺·PPᵢ complex structures showed a high degree of overall structural conservation (RMSD of 1.0-1.2 Å over 158-190 Cα atoms) except for conformational differences in loop 3. Notably, loops 1 and 2 adopted a similar conformation in all compared structures (Fig. 3a). The pterin group and the *p*ABA group of the ligands bound to Sul2/Sul3 or DHPS occupied the same spatial locations in all analyzed structures. The Sul2/Sul3 residues identified interacting with the pterin ring were also conserved in these representative DHPS enzymes (Figs. S5 and S7). Similarly, the Sul2/ Sul3 residues that coordinated Mg²⁺ and PPᵢ were also conserved in these DHPS enzymes (Figs. S5 and S7).

In contrast, we noticed a dramatic reorganization of the α6 helix that is involved in *p*ABA interactions in the Sul enzymes (Fig. 3b). Furthermore, the conserved Phe residue belonging to this helix (Phe178 in Sul1, Phe179 in Sul2, Phe177 in Sul3) and providing an additional hydrogen bond to *p*ABA through its backbone amide nitrogen is lacking in DHPS enzymes (Fig. 3b, c, d). The sidechain of this residue in the Sul enzyme structures was localized near the carboxylic acid group of *p*ABA and occupied a spatial location that would clash with the sulfonamide group and its *N*-acylation (i.e., the azole group of SMX) as revealed in the *Yp*DHPS·DHPP·SMX complex structure (Fig. 3e). The Sul enzymes lacked a Gly residue at the α6 helix present in DHPS (Gly191 in *Yp*DHPS). Another difference observed is that the conserved Leu residue in Sul enzymes (Leu180/Leu181/Leu179 in Sul1/ Sul2/Sul3, respectively) corresponded to a Lys residue in DHPS (Lys92 in *Yp*DHPS) (Fig. 3b, c, d, e). This residue appears packed against a conserved methionine (Met176/Met177/Met175 in Sul1/Sul2/Sul3); a smaller hydrophobic residue corresponding to a phenylalanine in DHPS (i.e., Phe188). These changes in the Sul enzymes may be responsible for residues between 179 to 181 to adopt a right-handed α-helical conformation; in contrast, only Gly189 adopts a left-handed α-helical conformation in the similar region of *Yp*DHPS. Since sulfas are *p*ABA structural analogs, we hypothesized that these observed alterations of the *p*ABA interaction region between Sul and DHPS may create a steric hindrance for sulfas that would affect their binding to the Sul enzymes.

Leveraging our structure of the Sul2·7,8-DHP·Mg²⁺·PPᵢ complex, we modeled 12 clinically relevant sulfonamides in the Sul2 active site through superposition with the *p*ABA region of 7,8-DHP (Fig. S8). This analysis showed that all 12 compounds would pose a steric clash with the Phe residue of α6 helix, including the most primitive sulfa sulfanilamide, whose structure is the closest mimic to *p*ABA. The modeling suggested that the sulfonamide nitrogen would be positioned 2.1 Å

distance from the Phe residue, resulting in an unfavorable clash. The modeling further suggested that the *N*-acyl substituents off the sulfonamide nitrogen that define the different compounds would exacerbate the clash with F179, at least using their energy-minimized conformations in our models.

As implied by this structural analysis, we suggested that the α6 helix Phe residue conserved in Sul enzymes is a key determinant of sulfa resistance. To test this hypothesis, we substituted this Phe to Gly in Sul1 and Sul3 and tested the kinetic properties of these recombinantly purified variants (Sul1$^{F178G}$ and Sul3$^{F177G}$) for utilization of *p*ABA to form 7,8-DHP, competition between *p*ABA and SMX, and consumption of SMX to form a pterin-SMX adduct. The Sul1$^{F178G}$ and Sul3$^{F177G}$ variants both showed $K_M$, $k_{cat}$ and catalytic efficiency parameters for *p*ABA similar to the wild-type (WT) Suls (Table 1), indicating that the catalytic properties of these Sul enzymes in the dihydropteroate synthase reaction remain unaltered by the substitution. In the competition experiment with *p*ABA, SMX inhibition of both the Sul1$^{F178G}$ and Sul3$^{F177G}$ variants was significantly increased as compared to WT Sul1 and Sul3, with a reduction in $K_i$ parameters of 243- and 90-fold, respectively. In the SMX utilization experiment, both the Sul1$^{F178G}$ and Sul3$^{F177G}$ variants showed increased preference for SMX as a substrate for the dihydropteroate synthase reaction, as demonstrated by increased $K_M$ (107- and 68-fold, respectively) and increased $k_{cat}/K_M$ (206- and 135-fold, respectively) as compared to the WT Sul1 and Sul3. Altogether, these results show that removal of the α6 Phe sidechain via Gly substitution in Sul1 and Sul3 made them enzymatically similar to sulfa-susceptible DHPS enzymes and indicate the key roles of this amino acid in the Sul *p*ABA-binding site for ligand discrimination.

### Sul enzymes differ from *Ec*DHPS in active site dynamics in response to ligand binding

We evaluated the conformational dynamics of the α6 helix and loop 3 elements of the Sul enzymes in response to ligand binding. Since Sul enzymes do not contain tryptophan residues, introduction of a single Trp residue to the α6 helix or loop 3 in the Sul enzymes would allow to evaluate the ligand-induced changes in these elements by intrinsic tryptophan fluorescence (ITF) (Fig. 4). Accordingly, we purified Sul1 and Sul3 variants carrying a Trp residue instead of the key Phe residue at the α6 helix (Phe178/Phe177 in Sul1/Sul3 respectively) (Fig. 4A, D). Similarly, we purified Sul1 and Sul3 variants with Trp replacing the Arg136 (Sul1) or Lys136 (Sul3) residues in loop 3. In addition we introduced individual Trp substitutions in Sul3 to the loop 3 residues Ala133, Thr135 or Val137 in Sul3.

All seven Sul1 and Sul3 variants showed dose-dependent DHPP or *p*ABA ligand-induced ITF changes (Fig. 5B, E, see also Fig. S9C). These results confirmed that the α6 helix and loop 3 of these Sul enzymes change conformation in response to co-substrate binding. Fits of the ITF titration data derived binding dissociation constants ($K_d$) for DHPP and *p*ABA ranging from 2 to 23 μM (Fig. 5 and Fig. S9). The close range of calculated $K_d$ values across tested Sul1 and Sul3 mutants suggested that interactions with co-substrates were not largely altered by the introduction of single Trp substitutions.

Using these ITF-sensitive Sul variants, we evaluated the potential conformational changes in these enzymes in response to SMX. Consistent with our enzymatic analysis showing significantly lower affinity of these enzymes to sulfas compared to *p*ABA, we did not observe any significant change in ITF in any of the Sul1 and Sul3 Trp variants upon titration of SMX (Fig. 4C, F and Fig. S9). Apparently, the α6 helix and loop 3 do not undergo significant conformational changes in the presence of SMX.

Next, we checked if the regions of *Ec*DHPS equivalent to α6 helix and loop 3 undergo conformational changes in response to the co-substrates or SMX. *Ec*DHPS contains a single Trp residue (Trp92) located on the opposite face of the (α/β)$_8$ barrel (Fig. S10A). We substituted this residue by a phenylalanine thus eliminating the source of

background ITF signal (Fig. S10B). In this variant we then replaced the Phe190 belonging to the α6 helix or Met148 belonging to the loop 3 by a tryptophan (Fig. S10A) and tested the resulting double mutants—*Ec*DHPS$^{W92F/F190W}$ and *Ec*DHPS$^{W92F/M148W}$—in the ITF assay. *Ec*DHPS$^{W92F/F190W}$ and *Ec*DHPS$^{W92F/M148W}$ showed ITF changes in DHPP concentration-dependent manner, resulting in calculated $K_d$ values of 8 and 13 μM, respectively (Fig. S10C). In contrast, we did not observe any detectable changes in ITF when these variants were tested by titration of *p*ABA in presence of excess of DHPP. Titration of *Ec*DHPS$^{W92F/F190W}$ and *Ec*DHPS$^{W92F/M148W}$ with SMX also produced no ITF change (Fig. S10C). Together, these data indicated that in contrast to Sul enzymes, no additional local conformational change occurs in these elements of *Ec*DHPS beyond those induced by DHPP binding.

### Expression of the individual sul genes in an *E. coli* strain confers pan-sulfa resistance

To assess the contribution of the *sul* genes and their derivatives on conferring sulfa resistance and fitness in *E. coli*, an unmarked, in-frame *folP* deletion was introduced in the parent strain, *E. coli* BW25113. In agreement with the previously reported thymidine-auxotrophic phenotype of the *E. coli* C600Δ*folP*::Kan$^R$ transposon mutant[52,56], deletion of the *folP* gene in BW25113 resulted in cells auxotrophic for thymidine. This deletion strain failed to grow in thymidine-limited Mueller-Hinton II medium but showed limited growth in thymidine-supplemented MHII medium (Fig. S11)[52,56]. Similarly, it also failed to grow on rich LB medium, which reportedly contains low amounts of thymidine[57] (Fig. S11). The growth of the Δ*folP* strain in MHII or LB medium was only partially restored by the addition of thymidine (Fig. S11). To confirm the role of *folP* in the thymidine auxotrophic phenotype, expression of WT *folP* in *trans* from a low-copy-number plasmid (pGDP2)[58] complemented the thymidine auxotrophy exhibited by the Δ*folP* mutant (Fig. S11).

Expression of the plasmid-borne *sul1*, *sul2*, or *sul3* genes fully restored growth of the *E. coli* Δ*folP* mutant under thymidine-limited conditions in a manner comparable to the WT parental strain and to the *folP*-complemented Δ*folP* strain (Fig. S12A). The expression of the Sul enzymes in these strains was confirmed by Western blot (Fig. S13). These data clearly demonstrated that the Sul enzymes can support the production of 7,8-DHP sufficient to fully restore the growth of a *folP*-null mutant. To assess the contribution of the individual Sul enzymes to sulfa resistance, we assessed the susceptibility of the Δ*folP* strain carrying the *sul1*, *sul2*, or *sul3* genes to 12 sulfas and co-trimoxazole. Expression of the Sul enzymes in the Δ*folP* mutant strain resulted in high levels of *pan*-sulfonamide resistance and increased resistance to co-trimoxazole when compared with the WT parental strain and to the *folP*-complemented Δ*folP* deletion strain (Table 2 and S2). We also utilized this *trans*-complementation strategy to confirm that none of the Trp substitutions in *Ec*DHPS, Sul1 or Sul3 designed for our ITF assay affect the growth or sulfa resistance of the Δ*folP* strain (Fig. S12F, G, H and Table S3).

### The α6 helix phenylalanine residue conserved across Sul enzymes is essential for their intrinsic sulfa resistance

Next, we assessed the role of the conserved Phe residue belonging to the α6 helix of the Sul enzymes in sulfa resistance and growth of *E. coli*. The corresponding codon in each of the plasmid-borne *sul* genes was deleted and we examined the impact of this mutation on sulfa resistance and complementation of the thymidine-auxotrophy of the Δ*folP* strain carrying such mutated *sul* genes. In case of the Sul2 and Sul3, deletion of the Phe residue compromised resistance to all 12 sulfa drugs tested and to co-trimoxazole (Table 2 and S2), while not impacting the growth (Fig. S12C, D) or enzyme expression levels (Fig. S13). In contrast, the deletion of Phe178 in Sul1 resulted in cells unable to grow on MHII agar and, as a result we were not able to measure the sulfa MICs. However, the strain carrying this Sul1 variant was able to

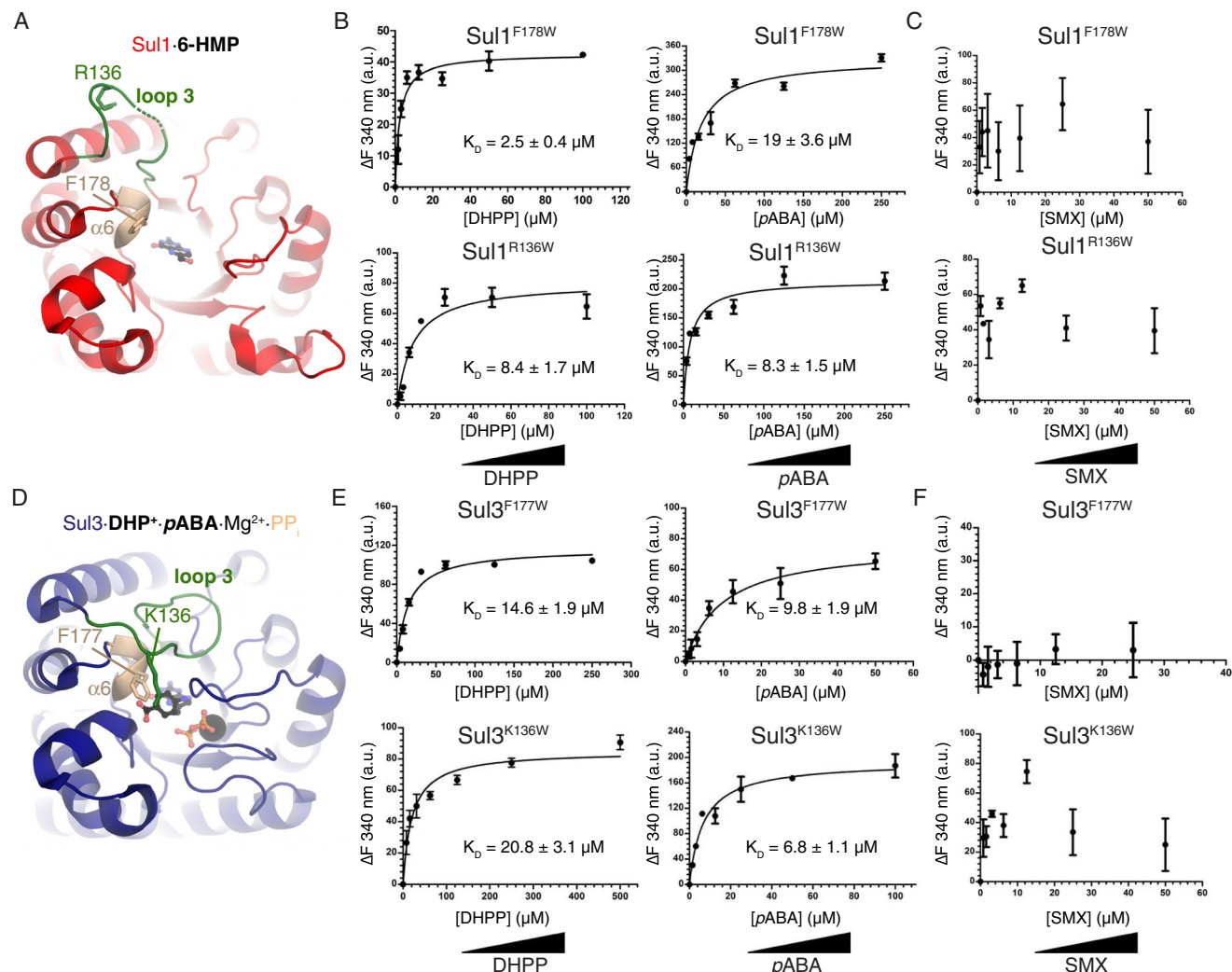

**Fig. 4 | Sul enzymes' α6 helix and loop 3 undergo conformational changes to binding of natural ligands but not to sulfas, as measured by ITF. A** Location of sites for introduction of Trp probes in the structure of Sul1. Titration of **B** DHPP and *p*ABA, and **C** SMX into Sul1$^{F178W}$ (top) and Sul1$^{R136W}$ (bottom). **D** Location of sites for introduction of Trp probes in the structure of Sul3. Titration of **E** DHPP and *p*ABA, and **F** SMX into Sul1$^{F177W}$ (top) and Sul1$^{K136W}$ (bottom). Vertical axes show substrate-induced fluorescence emission intensity change at 340 nm (error bars represent ±SD). Calculated $K_D$ values indicated under plots. Triplicate readings were averaged, and $K_D$ reported as (±SD).

grow, albeit slowly, in MHII broth (Fig. S13B). Of note, the Sul1 deletion did not adversely impact enzyme levels compared to the Δ *folP*-deletion strain expressing the WT Sul1 enzyme (Fig. S13), suggesting that deletion of this residue in Sul1 diminishes the enzyme's activity, limiting folate production and thus affecting the growth. Next, we substituted the conserved Phe residue in the three Sul enzymes with glycine and examined the effect of this substitution. In the case of Sul1, expression of this variant in the Δ*folP* strain did not adversely affect growth or the enzyme levels as compared to the Δ*folP* strains expressing the WT *sul1* gene (Figs. S12B and S13). This result is consistent with the notion that an amino acid at position 178 of Sul1 is essential for the dihydropteroate synthase reaction and, thus, growth. Similarly, substitution of Phe to Gly in Sul2 and Sul3 did not affect growth or enzyme expression levels (Fig. S12C, D). Substitution of Phe to a Gly residue in Sul1, Sul2, and Sul3 compromised resistance to all 12 sulfa drugs tested and to co-trimoxazole relative to the Δ*folP* strain expressing the WT enzymes (Table 2 and Table S2). The observed increase in susceptibility of the Sul Phe to Gly variants is consistent with our in vitro observations that drug binding to these variants is enhanced relative to WT. Thus, the Phe sidechain at this position in the Sul1, Sul2, and Sul3 enzymes is essential for conferring *pan*-sulfonamide resistance.

We also evaluated whether the conserved amino acid residue Leu, which is adjacent to conserved Phe residue on the α6 helix of Sul enzymes (L180 in Sul1, L181 in Sul2 and L179 in Sul3) contributes to sulfa resistance and growth of the Δ*folP* mutant. Mutation of this Leu residue to a negatively charged Glu residue in Sul1 or Sul3 compromised resistance to 12 sulfa drugs and to co-trimoxazole. Substitution of the conserved Leu with a positively charged Lys residue also compromised resistance to several sulfa drugs but to much lesser extents and did not impact co-trimoxazole resistance (Table 2 and S2). Notably, these substitutions of the Leu residue in Sul1 and Sul3 did not adversely impact growth (Fig. S12B, D) or enzyme expression levels (Fig. S13). These data suggest that the conserved Leu also plays an important role in reducing the affinity of Sul enzymes for sulfa molecules and confirms the general role of helix α6 in resistance.

### Adaptive evolution identifies sulfa-resistant FolP variants carrying a Sul-like Phe-Gly insertion

We were interested in the breadth of possible amino acid substitutions in the chromosomally-expressed *Ec*DHPS enzyme that would similarly confer resistance. We carried out four independent strain evolution experiments to assess if exposure of *E. coli* BW25113 to the prototypical sulfa drug, SAA, could generate specific mutations in *Ec*DHPS that

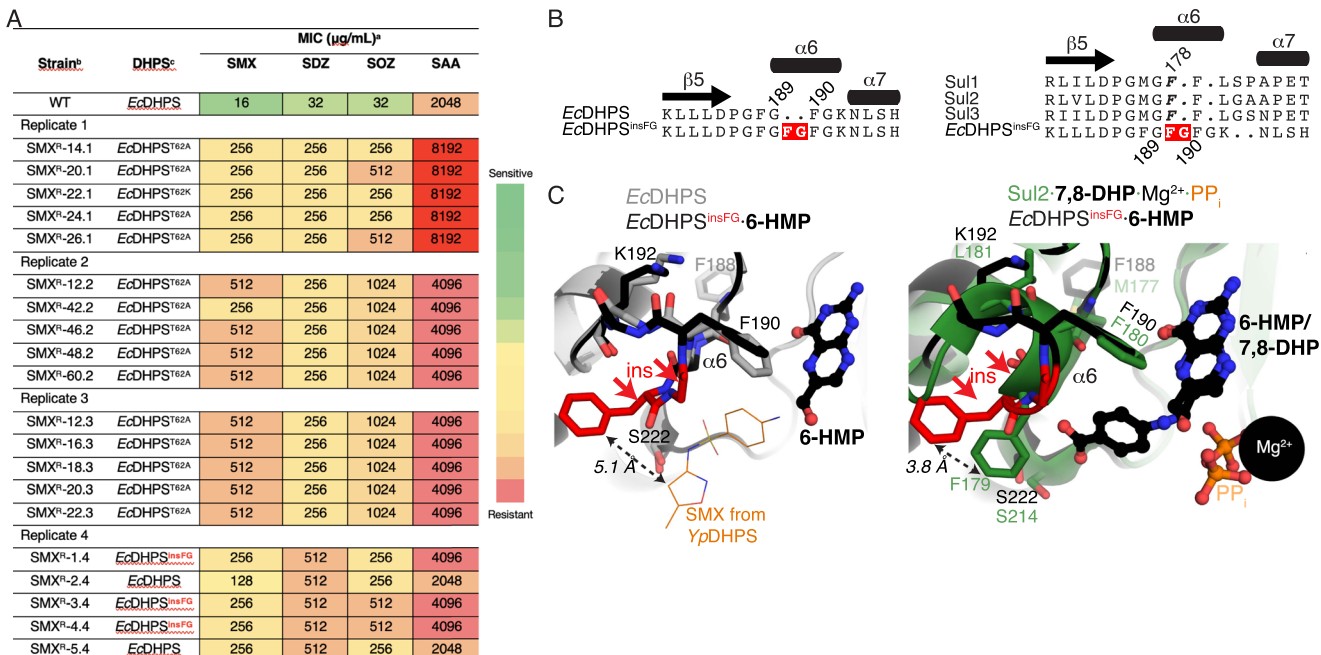

**Fig. 5 | Adaptive laboratory evolution recovers a *Ec*DHPS mutant with Sul-like modifications in α6. A** Sulfa resistance of the SMX-resistant mutants of *E. coli* BW25113 selected following a 7-day sulfanilamide exposure. **B** Sequence alignment of *Ec*DHPS^insFG^ insertion with *Ec*DHPS (left) and Sul enzymes (right), with insertion in *Ec*DHPS^insFG^ shaded red. At right, 178 refers to Phe178 from Sul1. **C** Left = Comparison of the crystal structures of the *Ec*DHPS^insFG^·6-HMP complex and *Ec*DHPS (PDB 1ajz). Superposed is the position of SMX from the *Yp*FolP·DHPP·SMX complex structure (PDB 3tzf), represented in thin lines. The distance between the inserted Phe and SMX is shown with a double-sided arrow. Right = Comparison of the crystal structures of the *Ec*DHPS^insFG^·6-HMP complex and the Sul2·7,8-DHP·Mg^2+^·PP_i complex. The distance between the inserted Phe residue in *Ec*DHPS^insFG^ and Phe179 in Sul2 is shown with a double-sided arrow. Insertion FG sequence is colored in red

and labeled with two red arrows in both panels of **C**. [a]Wild-type *E. coli* BW25113 was exposed to sulfanilamide (half the MIC; 1024 μg/mL) over 7 days and mutants resistant to 256 μg/mL of SMX were selected. Four independent in vitro laboratory evolution experiments were conducted (Replicates 1–4). [b]Results are for 5 randomly selected SMX-resistant (SMX^R) mutants from each independent trial are shown. Results for the *E. coli* BW25113 strain (WT) is provided for comparison purposes. For all strains and drugs tested, a minimum of 3 biological replicates were performed using the agar dilution method and MHII agar. [c]DHPS status of the indicated strains; the *folP* genes from each mutant was colony PCR amplified and sequenced to identify mutations in the DHPS coding region. *Ec*DHPS refers to the *E. coli* WT DHPS enzyme.

confer resistance similar to Sul enzymes. *E. coli* BW25113 was passaged daily over a 7-day period into MHII broth containing ½ the MIC of SAA (1024 μg/mL). On Days 1 and 7, cells were serially diluted and plated on MHII agar with or without super-MIC concentration of SMX (256 μg/mL). In all four independent trials, SAA exposure for one day did not result in the recovery of SMX-resistant (SMX^R) colonies, indicating that longer-term exposure was necessary. Similarly on Day 1, no SMX^R colonies were recovered for the drug-free and the vehicle control groups. 7-day exposure of *E. coli* to SAA led to the recovery of SMX^R colonies while no SMX^R colonies were recovered for the drug-free control and vehicle control groups. Twenty randomly selected SMX^R isolates (5 colonies from each trial) were subsequently picked and individually assessed for resistance to SAA and three additional sulfas (SMX, SDZ, SOZ). 18 isolates showed an increase in resistance to all four sulfa drugs while two isolates showed an increase in resistance to SMX, SDZ, SOZ, but not to SAA (Fig. 5A).

To assess the involvement of *Ec*DHPS in the observed increase in sulfa resistance, the 20 recovered SMX^R mutants were examined for mutations in the *folP* gene. Among all the mutations found in the coding region of *Ec*DHPS, the most frequently observed mutations involved Thr62 (located in loop 2, 15/20 isolates) (Fig. 5A and Fig. S5). Two SMX^R isolates were recovered in which no mutations were detected in the *folP* open reading frame, indicating the presence of mutation(s) elsewhere in the chromosome that confers sulfa resistance in these isolates. Three SMX^R isolates contained an insertion of six base pairs in the *folP* gene that encode a Phe-Gly insertion between residues 186 and 189 in WT *Ec*DHPS. Given the similarity of the modification identified in this *Ec*DHPS variant to the

corresponding features of Sul enzymes, we studied this mutant, dubbed *Ec*DHPS^insFG, in greater detail.

We determined the crystal structure of *Ec*DHPS^insFG^ at 2.73 Å. The crystal contained electron density in the active site corresponding only to only the pterin ring which we modeled as 6-HMP (Figs. 1 and 5C; Fig. S6). The structure revealed that the Phe-Gly insertion was localized to the α6 helix, extending it from two to four residues, and the Phe residue was positioned similarly to the position of the conserved Phe residues in the Sul enzymes (Fig. 5C). Comparison of the *Ec*DHPS^insFG^·6-HMP complex with the Sul2·7,8-DHP·Mg^2+^·PP_i complex showed that this inserted Phe was positioned 3.8 Å from the position of Phe179 in Sul2. Comparison of the *Ec*DHPS^insFG^·6-HMP complex with the *Yp*DHPS·DHPP·SMX complex showed that the inserted Phe was localized at 5.1 Å from the closest atom of the methoxazole group of superimposed SMX, while the backbone carbonyl of the inserted Phe was positioned at 4.2 Å to this atom of SMX. These observations indicate that the observed insertion of the Phe-Gly in the α6 helix of DHPS enzymes would occlude sulfa binding through steric repulsion, similarly to the mechanism we established for Sul enzymes.

To validate that the Phe-Gly insertion in *Ec*DHPS contributes to sulfa resistance, we cloned the *folP*_insFG gene into the pGDP2 plasmid and introduced the plasmid into the Δ*folP* strain to yield Δ*folP*/pGDP2::*folP*_insFG. This strain demonstrated increased resistance to all sulfas tested and co-trimoxazole compared to that of the Δ*folP* strain expressing WT *Ec*DHPS (Table S4). However, sulfa resistance levels of the *trans*-complemented strain remained lower than that of the evolved parent strain, suggesting that mutations elsewhere in the chromosome may also be contributing to sulfa resistance.

**Table 2 | Sulfa susceptibility of the *E. coli* Δ*folP* deletion expressing WT or mutated Sul enzymes in the *p*ABA interaction region[a]**

| Strain | Plasmid | DHPS[c] | Growth on MHII agar[d] | MIC (μg/mL)[b] | | |
|---|---|---|---|---|---|---|
| | | | | SMX | SDZ | SOZ |
| WT | None | *Ec*DHPS | G | 16 | 32 | 32 |
| Δ*folP* | None | None | NG | NG | NG | NG |
| Δ*folP* | pGDP2 | None | NG | NG | NG | NG |
| Δ*folP* | pGDP2 | *Ec*DHPS | G | 16 | 32 | 32 |
| Δ*folP* | pGDP2 | Sul1 | G | 2048 | 4096 | 2048 |
| Δ*folP* | pGDP2 | Sul1$^{F178G}$ | G | 16 | 32 | 64 |
| Δ*folP* | pGDP2 | Sul1$^{FΔ178}$ | NG | NG | NG | NG |
| Δ*folP* | pGDP2 | Sul1$^{L180K}$ | G | 2048 | 1024 | 2048 |
| Δ*folP* | pGDP2 | Sul1$^{L180E}$ | G | 64 | 64 | 2048 |
| Δ*folP* | pGDP2 | Sul2 | G | 2048 | 4096 | 2048 |
| Δ*folP* | pGDP2 | Sul2$^{F179G}$ | G | 32 | 64 | 128 |
| Δ*folP* | pGDP2 | Sul2$^{ΔF179}$ | G | 4 | 8 | 4 |
| Δ*folP* | pGDP2 | Sul3 | G | 2048 | 4096 | 2048 |
| Δ*folP* | pGDP2 | Sul3$^{F177G}$ | G | 16 | 32 | 128 |
| Δ*folP* | pGDP2 | Sul3$^{ΔF177}$ | G | 2 | 8 | 16 |
| Δ*folP* | pGDP2 | Sul3$^{L179K}$ | G | 2048 | 4096 | 2048 |
| Δ*folP* | pGDP2 | Sul3$^{L179E}$ | G | 64 | 128 | 256 |

Sensitive — Resistant (heatmap legend)

[a]The sulfa susceptibility of the WT *E. coli* BW25113 strain and the *E. coli* Δ*folP* strain carrying the indicated plasmids expressing *Ec*DHPS or Sul1, Sul2, Sul3 and Sul variants with the indicated amino acid substitutions or deletions is reported. Results for the WT *E. coli* BW25113 strain is provided for comparison purposes. Results for the plasmid-free and empty plasmid-carrying (pGDP2) *E. coli folP* deletion are provided to confirm the absence of thymidine in the MHII agar media. For all strains and drugs tested, a minimum of three biological replicates were performed using the agar dilution method and MHII agar.
[b]*SMX* sulfamethoxazole, *SDZ* sulfadiazine, *SOZ* sulfisoxazole.
[c]DHPS status of the indicated strains.
[d]*NG* no growth, or *G* growth on MHII agar.

Nonetheless, the *trans*-complementation with the *folP*$_{insFG}$ gene yielded WT levels of DHPS expression and complemented the thymidine auxotrophy (Figs. S12 and S13). Taken together, this data validated the key role of modification of the α6 helix residues in Sul/DHPS enzyme-mediated sulfa resistance. Finally, we utilized molecular dynamics to gauge whether variations in the sequence composition of the α6 region of Sul and *Ec*DHPS affect its flexibility. Simulations of the Sul2 WT, Sul2$^{F179G}$, *Ec*DHPS and *Ec*DHPS$^{insFG}$ *apo* enzymes revealed that both this point substitution in Sul2 and the insertion in *Ec*DHPS decreased α6's flexibility (root-mean-square-deviation, Fig. S15). These results suggested that Sul2$^{F179G}$ harbors a more organized binding site for sulfa drug binding, while *Ec*DHPS$^{insFG}$ provides a capability of this enzyme to adopt a specific conformation to occlude sulfa binding.

## Discussion

Comprehensive understanding of the mechanisms of antibiotic resistance is essential for development of novel antimicrobial therapies[59] In stark contrast to other resistance enzymes such as β-lactamases[59], detailed molecular understanding has been lacking in the case of plasmid-borne resistance to sulfa antibiotics. The sulfas currently approved for clinical use all share a common chemical scaffold while diverging in acylation at the 2-*N* position with an aromatic moiety (e.g., azole, oxazole, pyridine). Further alterations of the core scaffold have not been generated, to the best of our knowledge, while all approved sulfas are compromised by *sul*-mediated resistance, dramatically limiting their use.

Our results provide detailed analysis of the in vitro kinetic parameters for purified Sul enzymes, thus establishing the framework for evaluation of potential inhibitory compounds for these resistance enzymes. We unambiguously demonstrate that each of the Sul1, Sul2 and Sul3 enzymes harbor kinetic parameters comparable to DHPS enzymes as represented by *Ec*DHPS. Expression of an individual *sul* gene reversed the growth defect caused by the knockout of *folP*, indicating that Sul enzyme activity may be sufficient to compensate for sulfa inhibition of DHPS, at least in *E. coli*. Notably, while the kinetic properties of Sul1, Sul2 and Sul3 for utilization of *p*ABA were very similar, we observed a decrease in $K_M$ and $K_I$ parameters for Sul1, Sul2 and Sul3 enzymes for SMX.

Our data also clearly demonstrate that the Sul enzymes are capable of binding SMX and utilizing it as a substrate, condensing with the pterin moiety of DHPP to form a covalent adduct, which we detected by mass spectrometry. However, the catalytic efficiency for formation

of this dead-end adduct by Sul enzymes is more than 150-fold less than for the EcDHPS.

By introducing a single Trp residue at the appropriate structural elements of Sul and EcDHPS enzymes we were able to identify that conformational changes occur around the active site of these enzymes during catalysis via changes in the intrinsic tryptophan fluorescence signal. Our analysis revealed that the α6 helix and the nearby loop 3 in Sul enzymes undergo conformational changes in response to interactions with both pABA and DHPP. In contrast, these regions of EcDHPS change conformation only in response to DHPP binding. This is consistent with previous studies which showed that the rate-limiting step of DHPS activity is in release of PP$_i$ after DHPP binding, which then induces a fully formed pABA-binding site following an S$_N$1 ordered reaction mechanism. In the case of the Sul enzymes, we observed a similar kinetic trend, where binding of DHPP precluded the pABA-binding, however, the ITF data suggested that after this first step, the α6 helix and loop 3 continue to undergo conformational changes in response to pABA-binding. We hypothesize that this additional flexibility of the Sul enzyme active site may be important to occlude the binding of sulfas by positioning appropriate structural elements.

The determination of the crystal structures of the Sul1, Sul2 and Sul3 enzymes provided valuable molecular images of these enzymes. This structural data confirmed that the pABA-interaction region of their active sites includes the α6 helix and flexible loops as key elements involved in discrimination of sulfa molecules. More specifically, the Phe residue conserved in Sul but lacking in EcDHPS appears to intercept sulfa binding through a steric clash. Substitution of this Phe residue with a glycine dramatically improved the binding of SMX to each of the Sul enzymes while not compromising their catalytic activity. Accordingly, E. coli ΔfolP strain trans-complemented with any of the sul genes variants encoding this Phe-to-Gly mutation became susceptible to sulfa inhibition. In a reciprocal experiment, we characterized a lab-evolved sulfa-resistant E. coli strain carrying a folP variant encoding for EcDHPS possessing a Phe-Gly insertion in the equivalent region of its active site.

As further corroboration of the relevance of modifications in the α6 region of the pABA-interaction site of DHPS enzymes for resistance, sulfa-resistant clinical isolates of Neisseria meningitidis were shown to harbor a six bp insertion, coding for a Ser-Gly insertion in the corresponding region (Fig. S14)[60]. The same study demonstrated that deletion of this insertion in the N. meningitidis folP gene compromised sulfa resistance[60].

We also identify the Phe-Gly insertion in the putative α6 region of DHPS enzymes encoded by three E. coli genomes in the NCBI database (Genbank IDs: HAX1960291.1, EFG0716961.1 and EFI6540711.1). While the sulfa resistance status of these E. coli strains was not reported, our data allows us to predict that these strains would be sulfa-resistant. We also identify that the Gly-Phe-Phe sequence motif occurs naturally in hundreds of sequences deposited to NCBI and annotated as DHPS. This motif has been previously identified as a hallmark of Sul and Sul-like dihydropteroate synthases; some genes with this motif were shown to confer sulfa resistance when expressed in E. coli[51]. The significance of the natural circulation of this variant of these apparent DHPS enzymes for sulfa resistance remains unclear.

Although many studies have demonstrated that expression of the sul genes alone or in combination in E. coli confers sulfa resistance, few employed quantitative assays for MIC determination and of those that have, only a limited number of sulfa drugs have been tested[34,61–65]. This study thus comprehensively quantifies the contribution of plasmid-borne sul genes to several sulfas in a single strain of E. coli devoid of additional resistance genes. The results demonstrate that expression of the individual sul genes in E. coli results in high levels of resistance for all 12 sulfas tested. We observed a positive correlation between the MIC of the sulfas vs. WT

E. coli and E. coli ΔfolP/pGDP2::sul with the pK$_{a2}$ of the sulfonamide group[66]; this observation has been made before[67] and correlated with accumulation of the anionic form of the sulfas into bacterial cells[66]. In all instances, the MICs we measured surpassed the CLSI resistant MIC breakpoints, indicating that the sul genes alone are sufficient to compromise sulfa therapy[68]. These results have implications for cross-resistance to sulfas used either in animal husbandry or human health and suggest that efforts to segregate the use of sulfas into animal and human domains would be futile when it comes to sul-mediated resistance.

Finally, our structural and enzymatic data identified the chemical liability in all sulfa drugs which compromises their binding to Sul enzymes is the sulfonamide nitrogen itself plus the N-acylation substituents at this position. Thus, we posit that a core pharmacophore more closely resembling pABA is necessary to inhibit the Sul enzymes. This compound would need to have restricted size, evade the α6 Phe residue and/or leverage it for favorable hydrophobic interactions. There has been extensive research into the inhibition of DHPS enzymes through non-sulfonamide scaffolds[20,55,69–74], however, to the best of our knowledge, no campaigns have been conducted to discover Sul inhibitors, their pABA-binding site, or compounds capable of inhibiting both Sul and DHPS enzymes. Given the widespread dissemination the sul1 gene in particular, dual Sul1/DHPS inhibitors would be advantageous. The molecular and structural data we obtained the Sul enzymes can be utilized to initiate such efforts and provide new avenues for overcoming sulfa resistance.

## Methods

### Bacterial strains and growth conditions

Bacterial strains and plasmids used in this study are described in Table S5, primers used are listed in Table S6. Bacterial cells were cultured in Luria broth and Luria agar, Mueller-Hinton II broth and Mueller-Hinton II–agar with antibiotics, 200 μg/mL thymidine, and 40 μM folinic acid as necessary, at 37 °C. Plasmid pKOV was a gift from George Church (Addgene plasmid # 25769)[75]. In E. coli, plasmid pKOV and its derivatives were maintained with 25 μg/mL chloramphenicol at 30 °C, pGDP2 and its derivatives were maintained with 25 μg/mL kanamycin, pMCSG53 and its derivatives were maintained with 100 μg/mL ampicillin, pNIC-CH and its derivatives were maintained with 50 μg/mL kanamycin.

### DNA Methods

Standard protocols were used for restriction endonuclease digestion, ligation, transformation, and agarose gel electrophoresis[76]. Plasmid and chromosomal DNA were prepared as before[77]. Dephosphorylation and ligation into the gene replacement vector pKOV was carried out using the Rapid DNA Dephos & Ligation kit (Roche). CaCl$_2$-competent E. coli cells were prepared as described[78]. Electrocompetent E. coli cells were prepared using a method by New England Biolabs (Ipswich, MA). Oligonucleotide synthesis was carried out by Integrated DNA Technologies (Coralville, IA). In several instances, genes were designed in silico, synthesized and/or mutated and cloned into the required plasmid at BioBasic Inc (Markham, Canada). For in vitro protein characterization and crystallization, cDNA for sul1, sul2, sul3 were purchased from IDT DNA Technologies as Gblock gene fragments (1-269 amino acids for Sul1, residues 1-271 amino acids for Sul2 and 1–242 amino acids for Sul3) and were codon optimized for expression in E. coli. All primers were purchased from IDT DNA technologies. cDNA for sul2, sul3 were cloned into pMCSG53 (N-terminal 6x histidine tag cleavable by TEV protease). cDNA for sul1 was cloned into pNIC-CH (non-cleavable His$_6$ tag). E. coli K12 DHPS (folP, 1-282 amino acids) and E. coli K12 HPPK (folK) were cloned into pMCSG53 for enzymatic kinetic studies. Site-directed mutants were prepared using a method modified from the QuickChange protocol (Stratagene, La Jolla, CA).

## Protein purification

*E. coli* BL21(DE3) Gold competent cells were transformed with pMCSG53 or pNIC-CH expression plasmids harboring *sul1*/*sul2*/*sul3* or *folP*. 20 mL of overnight culture (approx. 16 h growth time) in LB was diluted into 1 L of LB containing selected antibiotics (kanamycin for pNIC-CH plasmids, ampicillin for pMCSG53 plasmids) and grown at 37 °C with shaking. Expression was induced with 0.5 mM IPTG at 17 °C when $OD_{600}$ reached 0.8 units and allowed to grow overnight for 16–18 h. Overnight cultures were collected by centrifugation at 7000 g then resuspended in binding buffer [pH 7.5, 100 mM HEPES, 500 mM NaCl, 5 mM imidazole, and 5% glycerol ($v/v$)] and lysed by sonication. Cell debris was removed by centrifugation at $20,000 \times g$. The soluble fraction was purified by batch-binding to a Ni-NTA resin, washed with wash buffer [pH 7.5, 100 mM HEPES, 500 mM NaCl, 30 mM imidazole, and 5% glycerol ($v/v$)], and protein was eluted with elution buffer [pH 7.5, 100 mM HEPES, 500 mM NaCl, 250 mM imidazole, and 5% glycerol ($v/v$)]. The $His_6$-tagged protein was then subjected to overnight (approx. 16 h) cleavage using 50 µg of TEV protease per mg of $His_6$-tagged protein and simultaneously dialyzed overnight against a buffer containing no imidazole. The $His_6$-tag and TEV were removed by applying to a Ni-NTA column again and flowthrough was collected. For plasmid constructs with non-cleavable $His_6$-tag, the TEV protease cleavage step was skipped. The final proteins were dialyzed with a minimum of $2 \times 2$ L dilution cycles in 10 mM HEPES at pH 7.5 with 300 mM NaCl, 0.5 mM TCEP for crystallization or in 50 mM HEPES at pH 7.5 with 300 mM NaCl, 0.5 mM TCEP for kinetics. Protein purity was analyzed by SDS-PAGE and mass spectrometry. The purified proteins were also subjected to size exclusion chromatography (Superdex 200 16/60) analysis for determination of their oligomeric state. The proteins were concentrated using a Vivaspin concentrator (GE Healthcare) and passed through a 0.2 µm Ultrafree-MC centrifugal filter (Millipore) before storing in aliquots at −80 °C.

## In vitro dihydropteroate synthase activity assay

DHPP was synthesized based on previous methods[70] and its purity was validated by mass spectrometry. *E. coli* K12 HPPK (6-hydromethyl-7,8-dihydropterin pyrophosphokinase) with N-terminal $His_6$ tag was purified using Ni-NTA chromatography as described above. 6-HMP (Schirks Laboratories, Switzerland) was subjected to enzymatic conversion at 37 °C for 1 h with shaking using 5 mM ATP, 30 mg/mL HPPK, 10 mM $MgCl_2$, 3% DMSO in 50 mM HEPES pH 7.5. The reaction mixture was filtered through a 3 kDa MWCO filter. The filtrate containing the product DHPP, traces of ATP and unconverted substrate 6-HMP was analyzed by mass spectrometry; 99% conversion was achieved as observed by depletion of 6-HMP. The purified DHPP was droplet frozen in aliquots and stored at −80 °C. *p*ABA was purchased from Sigma. Dihydropteroate synthase activity by DHPS/Sul enzymes was monitored spectrophotometrically by a coupled Malachite green assay[79]. Malachite green solution was prepared in concentrated sulfuric acid (1.1 g in 150 mL conc. $H_2SO_4$ to total 1 L with water). Ammonium molybdate (Fluka) was prepared as a 7.5% (w/v) solution in water. For the assay, fresh Malachite green reaction (MGR) mix was prepared by mixing 4 mL Malachite green solution, 2 mL 7.5% (w/v) ammonium molybdate, 80 µL 11% Tween 20. The free pyrophosphate released in the DHPS reaction was cleaved to ortho-phosphate using inorganic *E. coli* K12 pyrophosphatase (NEB) in the enzymatic reaction (0.1 U/mL). The enzyme reaction was carried out in 96-well format containing the substrates DHPP and *p*ABA/SMX, 1 µM *Ec*DHPS or Sul enzyme, 10 mM $MgCl_2$, 2% DMSO, 50 mM HEPES pH 7.5, 0.01 U of inorganic *E. coli* pyrophosphatase, all in a 100 µL reaction. The 96-well plate reaction was incubated on a benchtop temperature-controlled plate shaker (ELMI) at 37 °C for 20 min at 500 rpm. Ortho-phosphate was color-imetrically detected by adding 25% of total well reaction volume of fresh MGR mix, shaking for 2 min and monitoring absorbance. Absorbance values at 630 nm were instantly monitored on an Infinity

MPlex Tecan plate reader with linear shaking of 60 s at 2.5 mm amplitude and 25 pixel readings per well read-out. Data from three independent biological replicates were averaged. Kinetic parameters ($K_M$) were determined by non-linear least-squares fitting of the data to the Michaelis-Menten equation, using GraphPad Prism v5.0 software (GraphPad Software, U.S.).

For the bisubstrate enzyme reaction, two different experiments were performed. For DHPP kinetics, DHPP concentrations were varied in the range of 0–500 µM in the presence of 10-fold excess of *p*ABA (200 µM). For *p*ABA kinetics, *p*ABA concentrations in the range of 0–200 µM were used in presence of the DHPP concentration at 20-fold excess of its $K_d$ at 200 µM. For the SMX kinetics, SMX concentrations were in the range of 0–20 mM for Sul enzymes and 0–80 µM for Sul variants and *Ec*DHPS, while the DHPP concentration was set at 20-fold excess of its $K_d$ at 200 µM. For the $K_i$ determination of SMX, three different SMX concentrations were tested, with varying *p*ABA concentrations (0–200 µM) and excess DHPP at 200 µM. $K_i$ for each SMX concentration was calculated using the formula $K_i = [I] / [(K_{Mobs}/K_M) - 1]$ and averaged. Here, [I] is the SMX concentration used, $K_{Mobs}$ is the observed $K_M$ at the respective inhibitor SMX concentration and $K_M$ is at zero inhibitor SMX concentration.

## Crystallization, X-ray data collection, and structure solution

All crystallization was performed at room temperature of 21 °C by the sitting drop vapor diffusion method with 0.6 µL protein and/or protein:ligand substrate mix plus 0.6 µL reservoir solution, using a STP Labtech Mosquito robot. *p*ABA was 20 mM stock solution in water. DHPP was 5 mM stock solution in 50 mM HEPES pH 7.5, 10 mM magnesium chloride ($MgCl_2$), 3% ($v/v$) dimethyl sulfoxide (DMSO). For the Sul1·6-HMP complex crystal, 1 mM solution of protein was incubated for 2 h with 2.5 mM DHPP at 4 °C then passed through a 0.2 µm Ultrafree-MC centrifugal filter. The reservoir solution was 25% (w/v) PEG 5 K MME, 0.2 M ammonium sulfate, 0.1 M Tris (pH 8.5), 1% (w/v) tri-isobutylmethylphosphonium tosylate. The crystal was cryoprotected by transferring to well solution with 10% ($v/v$) 2-methyl-2,4-pentanediol followed by paratone oil. The Sul2 *apo* crystal was grown in reservoir solution 1.6 M ammonium sulfate, 2% ($v/v$) 1,6-hexanediol, 0.1 M HEPES (pH 7.5) and cryoprotected either by transferring in reservoir solution with 2% ($v/v$) PEG 200 followed by paratone or using only paratone oil respectively. The Sul2·7,8-DHP·$Mg^{2+}$·$PP_i$ complex crystal was grown by incubating a 1 mM solution of protein with 2 mM DHPP for 30 min at 4 °C followed by adding 2 mM *p*ABA and further incubating for 2 h at 4 °C then passed through a 0.2 µm Ultrafree-MC centrifugal filter. The reservoir solution for this crystal was 1.6 M ammonium sulfate, 2% ($v/v$) hexanediol, 0.1 M HEPES pH 7.5. The crystal was cryoprotected by transferring to well solution with paratone oil. WT Sul3 failed to form crystals; the double mutant E142A + E143A crystallized (surface entropy reduction approach[80]). For the Sul3 apoenzyme complex, Sul3.E142A + E143A crystals were grown in reservoir solution of 2 M ammonium sulfate, 5% (w/v) isopropanol and cryoprotected by transferring in reservoir solution with 2.5% ($v/v$) glycerol followed by paratone. For the Sul3·6-HMP complex, a 1 mM solution of Sul3 and 2 mM DHPP was incubated for 1 h at 4 °C, then grown in reservoir solution 20% (w/v) PEG3350, 0.1 M calcium chloride and 0.05 magnesium chloride. The crystal was cryoprotected by transferring to reservoir solution containing 0.5% trehalose, 0.5% glycerol and paratone oil. For the Sul3·7,8-DHP·*p*ABA·$Mg^{2+}$·$PP_i$ complex, a 1 mM solution of Sul3.E142A + E143A was incubated for 30 min at 4 °C with 2.5 mM DHPP followed by 5 mM *p*ABA for a further 2 h at 4 °C and passed through a 0.2 µm Ultrafree-MC centrifugal filter. Crystals were grown in reservoir solution 2 M ammonium sulfate and 5% (w/v) isopropanol. The crystal was cryoprotected by transferring to a reservoir solution containing 12% ($v/v$) glycerol, 5% ($v/v$) trehalose followed by paratone. The *Ec*DHPS$^{insFG}$·6-HMP crystal was grown by mixing a 1 mM solution of the protein with 2 mM DHPPP for 40 min, then 2 mM *p*ABA for 40 min

and passed through a 0.2 μm Ultrafree-MC centrifugal filter. Crystals were grown in reservoir solution 0.1 M Tris pH 8.5, 0.2 M MgCl₂ and 25% (w/v) PEG3350, then cryoprotected in paratone. Prior to data collection, crystals were flash frozen in liquid nitrogen. X-ray diffraction data at 100 K was collected at the beamline 19-ID, Structural Biology Center, Advanced Photon Source, Argonne National Laboratory (wavelengths, Å, as follows: Sul1·6-HMP–0.97918, Sul2 apoenzyme–0.97951; Sul2·7,8-DHP·Mg²⁺·PPᵢ–0.97918; Sul3 apoenzyme–0.978; Sul3·6-HMP–0.97918; Sul3·7,8-DHP·pABA·Mg²⁺·PPᵢ–0.978; $Ec$DHPS$^{insFG}$·6-HMP–0.97913. All diffraction data were processed using HKL3000[81]. Molecular Replacement was utilized to solve each structure, using the structure of $Ec$DHPS (PDB 1AJO[82] as the search model, or the apoenzyme version of each enzyme, with Phenix.refine[83]. All model building and refinement were performed using Phenix.-autobuild, Phenix.refine and Coot[84]. Electron density features in the active sites of Sul1 and $Ec$DHPS$^{insFG}$ crystals was modeled as 6-HMP, likely resulting from the loss of PPᵢ from the DHPP in the crystallization mixture. The positions of all ligands were evaluated using simulated annealing omit maps using Phenix.refine. The occupancy of DHP⁺ in the Sul3·DHP⁺·pABA·Mg²⁺·PPᵢ complex was refined to 0.80. All geometry was evaluated using Phenix.molprobity and the wwPDB validation server. Ramachandran statistics are as follows (favored/allowed/outliers, %): Sul1·6-HMP–97.1/2.9/0; Sul2 apoenzyme–98.1/1.9/0; Sul2·7,8-DHP·Mg²⁺·PPᵢ–98.1/1.9/0; Sul3 apoenzyme–95.0/5.0/0; Sul3·6-HMP–94.2/5.8/0; Sul3·7,8-DHP·pABA·Mg²⁺·PPᵢ–98.1/1.9/0; $Ec$DHPS$^{insFG}$·6-HMP–95.6/4.4/0. Atomic coordinates have been deposited in the Protein Data Bank with accession codes 7S2I, 7S2J, 7S2K, 7S2L, 7S2M, 7TQ1 and 8SCD. Production of figures and analysis was performed using The PyMOL Molecular Graphics System, Version 2.4.0 Schrödinger, LLC. Sequence alignment was performed by the Clustal Omega server with manual adjustments based on structural superpositions, and figure was produced using the ESPript server.

### Intrinsic tryptophan fluorescence (ITF)

Site-directed mutagenesis (QuikChange) was used to generate Trp mutations in Sul1, Sul3 and $Ec$DHPS in plasmids pMCSG53/pNIC-CH. The experiment was performed in a black, opaque 96-well microtiter plate (ThermoScientific) in a 100 μL total reaction volume containing 1 μM of enzyme in 50 mM HEPES pH 7.5, 10 mM MgCl₂, 2% DMSO. Serial dilutions of substrate DHPP (5 mM stock), pABA (0.5 mM stock), SMX (200 mM stock solution in DMSO), SDZ (200 mM stock solution in DMSO) were used. According to the S$_N$1 reaction mechanism of Suls, to investigate the ITF change in response to pABA, we performed the ligand titration in presence of saturating excess of DHPP. To account for background signal from the ligand, control buffer titrations with the ligand alone were performed in triplicate for each titration point. The plate was allowed to incubate for 30 min at 37 °C on a bench plate shaker (ELMI BioSciences). ITF spectroscopy experiments were recorded on an Infinite MPlex Tecan plate reader, settings: top-read, λex 295 nm, λem 340 nm, photomultiplier gain 40, 60 s shaking prior to endpoint read and 25 pixel points/well read-out. ITF spectra were recorded at 295 nm excitation wavelength and emission scan recorded. The background fluorescence quenching caused by protein dilution with the buffer was monitored by running parallel buffer control titrations. The averaged negative control data was subtracted from the ligand titration data for each titration point to remove the background signal from the ligand ($F$). The value of $F$ before the start of titration (no ligand added), where the protein is unsaturated is $F_o$. The value of ($F$) at each ligand concentration was subtracted from ($F_0$) giving $\Delta F$ for each ligand concentration. Readings for each experiment were recorded in triplicate and averaged. Data were analyzed by plotted as fluorescence intensity changes $\Delta F$ against corresponding ligand concentrations and fit to a one-site binding equation (Chergoff Hill's plot) to generate $K_D$ and

$B_{max}$. $K_D$ is the equilibrium dissociation constant and Bmax is the maximum specific binding. Graphpad Prism v5.0 software (GraphPad Software, U.S.) was used for curve fitting.

### Mass spectrometry for identification of pterin-SMX covalent adduct

Reactions were carried out in a 96-well format containing the substrates, DHPP and SMX, along with 1 μM $Ec$DHPS or Sul1, 10 mM MgCl₂, 2% DMSO, 50 mM HEPES pH 7.5, all in a 100 μL reaction volume. The 96-well plate reaction was incubated on a benchtop temperature-controlled plate shaker (ELMI) at 37 °C for 20 min at 500 rpm. For $Ec$DHPS, the SMX concentrations (0, 4, 8, 20, 500 μM) were chosen to be in the range of its $K_M$ as determined by MG assay (7.7 μM; range above and below this $K_M$). Likewise, for Sul1, the range of SMX concentrations chosen were 0, 20, 500, 1000, 2500 μM to be in the range of its $K_M$ (1000 μM; range above and below this $K_M$). The DHPP concentration was set at 20-fold excess of its $K_D$ as determined by ITF, 200 μM, in all these experiments. A control plate with only DHPP (200 μM) and at the varying [SMX] were also set up in parallel under similar experimental conditions. For mass spectrometry, chromatography was carried out on a Thermo Scientific Hypersil Gold C18 column (50 mm × 2.1 mm, 1.9 μm) (Thermo Fisher Scientific, Waltham, MA) equipped with a guard column, using a Thermo Scientific Ultimate 3000 UHPLC (Thermo Fisher Scientific, Waltham, MA). The column temperature was 40 °C and the flow rate was 300 μL·min⁻¹. The eluents used were water (A) and acetonitrile (B), and both eluents contained 0.1% formic acid. The gradient started at 5% B and was held for 1 min, followed by a linear gradient to 98% B over 4 min, then a hold at 98% B for 5 min, a return to 5% B over 0.5 min, and finally a re-equilibration under the initial conditions of 5% B for 4.5 min (total runtime 15 min). Liquid samples (10 μL) were injected using an Ultimate 3000 UHPLC autosampler, with autosampler temperature of 8 °C. Compounds were detected and quantified using a Q-Exactive Orbitrap mass spectrometer (Thermo Fisher Scientific) equipped with a Heated Electrospray Ionization (HESI II) probe, operating in positive ionization mode. Mass spectra were acquired over an m/z range from 200 to 500 with the mass resolution set to 140 k, and common setting parameters were as follows; AGC Target: 3E6, max injection time 100 ms, spray voltage 3.5 kV, capillary temperature 320 °C, sheath gas 25, aux gas 5, spare gas 2, and s-lens RF level 55. Extracted ion chromatograms were generated for sulfamethoxazole (m/z 254.0594), reduced SMX-pterin (m/z 461.1242) or oxidized SMX-pterin (m/z 429.1090) using a 5 ppm error window in Xcalibur Qual Browser (Thermo Fisher Scientific). Calibration standard solutions of SMX were prepared from successive dilutions of a purchased SMX standard using the same reaction buffer used in the enzymatic reactions.

### Construction of E. coli BW25113 ΔfolP mutant

To generate an in-frame $folP$ gene deletion in WT $E.$ $coli$ BW25113, -1 kb fragments upstream and downstream of $folP$ were PCR amplified and cloned into plasmid pUC19, and then subsequently subcloned into pKOV. The $folP$ upstream fragment was PCR amplified using Δ$folP$ Up For and Up Rev and the $folP$ downstream fragment was amplified using Δ$folP$ Dn For and Δ$folP$ Dn Rev (Table S6). The 1 kb upstream and downstream fragments were PCR amplified from the chromosome of $E.$ $coli$ BW25113 parent strain in two, separate 50 μL mixtures that contained 1 μg of BW25113 chromosomal DNA, 0.6 μM of the appropriate primer set, 0.2 mM of each dNTP, 1x Phusion HF buffer, 5% (v/v) DMSO, and 1 unit (U) of Phusion DNA polymerase (Finnzymes, New England Biolabs, Pickering, ON, Canada). The mixture was heated for 30 s at 98 °C, followed by 30 cycles of 30 s at 98 °C, 30 s at 65 °C, 30 s at 72 °C, concluding with 7 min at 72 °C. The PCR products were subsequently gel-purified and digested with HindIII and XbaI or XbaI and BamHI, as appropriate, and separately cloned into appropriately restricted pUC19, yielding plasmids pMJF200 (upstream fragment)

and pMJF201 (downstream fragment). Both plasmids were individually introduced into DH5α, and transformants were selected on ampicillin 100 µg/mL. Plasmid DNA was recovered and sequenced to confirm the absence of mutations in the cloned fragments, the upstream fragment was excised from pMJF200 by digestion with HindIII-XbaI and was cloned into HindIII-XbaI restricted pMJF201, yielding pMJF202.

To generate the *folP* deletion construct in pKOV, the *folP* upstream and downstream fragments were PCR amplified from the purified plasmid, pMJF202 with the primers ΔfolP For NotI and DN Rev BamHI (Table S6) in a 50 µL mixture that contained 10 ng of template, 0.6 µM of the appropriate primer, 0.2 mM of each dNTP, 1x Phusion HF buffer, 5% (v/v) DMSO, and 1 U of Phusion polymerase (New England Biolabs). The mixture was heated for 30 s at 98 °C, followed by 30 cycles of 30 s at 98 °C, 30 s at 65.0 °C, 1 min at 72 °C, concluding with 7 min at 72 °C. The resulting ~ 2 kb PCR fragment was gel-purified, digested with NotI and BamHI, and cloned into dephosphorylated NotI-BamHI restricted, pKOV, to yield pKOV::ΔfolP (pMJF203). pMJF203 DNA was transformed into DH5α, and transformants selected on 25 µg/mL of chloramphenicol at 30 °C. Plasmid DNA was recovered and then sequenced. pMJF203 was electroporated into *E. coli* Keio BW25113 and allowed to recover for 1 h at 30 °C. Electroporants were then plated on prewarmed LB agar plates containing 25 µg/mL of chloramphenicol, 200 µg/mL of thymidine, and 40 µM of folinic acid and incubated overnight at 42 °C. From the 42 °C plate, 4-5 colonies were picked and suspended into 1 mL of LB with no NaCl, serially diluted, and immediately plated on 5% w/v sucrose, L-agar plates without NaCl, 200 µg/mL thymidine, and 40 µM folinic acid and incubated at 30 °C overnight. The next day, colonies were replica plated onto LB agar containing 5% w/v sucrose, no NaCl, 200 µg/mL thymidine, and 40 µM folinic acid and on MHII agar (no thymidine) to screen for loss of DHPS activity, and LB agar containing thymidine, folinic acid and chloramphenicol to confirm loss of the replacement vector. Of the 250 colonies that were replica plated, 2 colonies failed to grow on MHII agar, but not the sucrose containing LB agar plates. Deletion of the *folP* genes was confirmed using colony PCR with primers ΔfolP. For NotI and Dn Rev BamHI. A 10 µL colony PCR reaction mixture contained 2 µL of the chromosomal DNA solution as the template, 0.6 µM of each of primer ΔfolP For NotI and Dn Rev BamHI, 0.2 mM of each dNTP, 1x Thermopol buffer, 5% (v/v) DMSO, and 1 U of Taq DNA Polymerase (New England Biolabs, Whitby, ON, Canada). The mixture was heated for 3 min at 95 °C, followed by 35 cycles of 30 s at 95 °C, 45 s at 61 °C, 3:30 min at 72 °C, concluding with 5 min at 72 °C.

### Growth curve assay
*E. coli* strains were cultured overnight in MHII broth supplemented, as required, with 40 µM folinic acid, 200 µg/mL thymidine and/or 25 µg/mL kanamycin. Overnight cultures were diluted 1/10 in fresh MHII broth, pelleted by centrifugation ($5000 \times g$, 1 min), washed twice with MHII broth and then standardized to an $OD_{600}$ of 0.1. Using Corning Costar, Clear, 96-well round bottom microplates, bacterial cells were diluted in fresh MHII broth to a final $OD_{600}$ nm of 0.05. Growth was monitored at an $OD_{600}$, every 20 min for 24 h, with a MultiSkan GO Plate Reader and SKANIT Software CF (Thermo Fisher Scientific). In all instances, media only controls were carried out to ensure the absence of contamination and background absorbance at 600 nm.

### Sulfonamide antibiotic susceptibility testing
The susceptibilities of various *E. coli* strains to sulfonamide antibiotics were assessed using the agar dilution method[85] with the exception that MHII agar (Sigma) plates were used, as this agar is specifically formulated to have low levels of thymine and thymidine content, both of which are known to antagonize the action of sulfonamides. Antibiotics used were: sulfamethoxazole (SMX) (Sigma-Aldrich), 50 mg/mL in DMSO; sulfadiazine (SDZ) (Sigma-Aldrich), 100 mg/mL in 1 M NaOH; sulfanilamide (SAA) (Sigma-Aldrich), 50 mg/mL in acetone;

sulfisoxazole (SOZ) (TCI America) 50 mg/mL on acetone; sulfathiazole (STZ) (Sigma-Aldrich) 50 mg/mL in acetone: sulfapyridine (SPY) (Sigma-Aldrich) 40 mg/mL in 0.5 M NaOH; sulfamerazine (SMRZ) (AK Scientific), 50 mg/mL in DMSO; sulfamethazine (SMZ) (Alfa Aesar), 50 mg/mL in DMSO; sulfameter (SMT) (Sigma-Aldrich) 100 mg/mL in DMSO; sulfacetamide (SAD) (Sigma-Aldrich) 100 mg/mL in DMSO; sulfaquinoxaline (SQX) (Sigma-Aldrich), 50 mg/mL in DMSO; sulfadimethoxine (SDT) (Sigma-Aldrich), 100 mg/m: in 1 N NaOH. Freshly streaked *E. coli* colonies were resuspended in sterile 0.9% NaCl solution and the turbidity of the suspension adjusted to that of a McFarland Standard of 0.5 using a Sensitire Nephelometer (Thermo Fisher Scientific) calibrated to a McFarland 0.5 $BaSO_4$ standard. The bacterial suspension was diluted 1:10 into a well of a sterile 96-well microtitre plate by pipetting 10 µL into a well containing 90 µL of sterile saline. A 48-pin replicator with 1.5 mm pins was used to deliver the final inoculum of 1 µL (~$10^4$ CFU/mL) onto MHII agar plates containing various concentrations of either SMX, SDZ, SAA, SOZ STZ, SPY, SMRZ, SMZ, SMT, SAD, SQX, or SDT. The MIC for SMT at 8192 µg/mL and for SQX at ≥2048 µg/mL could not be determined as the sulfa drug precipitated out of solution at these concentrations. Inoculated sulfa-agar plates were incubated 37 °C for 18 h and the minimum inhibitory concentration (MIC) was determined to be the lowest concentration of drug that inhibited bacterial growth. In all instances, a drug-free control plate was included to ensure bacterial growth. The ΔfolP deletion strain, which is auxotrophic for thymidine and thus cannot grow on MHII agar was included as a negative control (no growth control). MIC values for co-trimoxazole were determined by E-test using trimethoprim-sulfamethoxazole (1:19) MIC test strips (Liofilchem, Italy). The MICs were read according to the E-test reading guide for TMP-SMX where the inhibition ellipse intersects the strip at 80% inhibition.

### Whole-cell protein extracts and western immunoblotting
Overnight cultures of *E. coli* strains grown in MHII-broth supplemented, when necessary, with 40 µM folinic acid, 200 µg/mL thymidine, and kanamycin 25 µg/mL, were subcultured (1:49) in MHII broth and incubated at 37 °C and shaking at 200 rpm until $OD_{600}$ of 0.5-0.6. Cells were then standardized to an $OD_{600}$ of 0.5 and pelleted by centrifugation at 5000 g. Pellets were resuspended in 200 µL 1x phosphate buffered saline (PBS) and 200 µL 2x RedMix. Samples were heated at 95 °C for 5 min and sonicated for 25 s at 30% amplitude. Whole-cell protein extracts were then separated on a 12% Mini-PROTEAN TGX Stain-Free protein gels and electroblotted onto a PVDF membrane using a BioRad Trans-BlotR Turbo™ Transfer System according to manufacturer's instructions. Blotted membranes were subsequently incubated in PBS containing 0.1% (v/v) Tween 20 (PBST) and 10% (w/v) skim milk for 1 h. Following two 5-min washes with PBST, the membranes were incubated for 60 min with a primary mouse monoclonal Anti-FLAG M2 antibody (1:5000 dilution; Sigma-Aldrich), in PBST containing 1% (w/v) bovine serum albumin (BSA) (Sigma-Aldrich) and then washed four times for 10 min each time with PBST. A secondary polyclonal Goat Anti-Mouse IgG H&L (HRP) antibody (1:5000 dilution; Abcam, Massachusetts) in PBST containing 1% BSA was added to the membranes and incubated for 1 h, and subsequently washed four times for 10 min each time with PBST. All incubations and washings in the immunoblot procedure were carried out at room temperature with gentle agitation. Blots were developed by using the Clarity Western ECL Substrate (Biorad), according to the manufacturer's instructions and the blots were visualized using DNR Bio-Imaging Systems MicroChemi 4.2 imaging system with GelCapture (DNR Bio-Imaging) software following 15 s exposure.

### Adaptative laboratory evolution
100 µL of an overnight culture of *E. coli* Keio BW25113 in MHII broth was transferred into 10 mL of fresh MHII-broth containing either

1024 µg/mL (1/2 MIC) of SAA or no drug. Acetone, which was used to solubilize SAA, was included as a vehicle control (at similar concentrations as used in the SAA-exposed cells). Following a 24 h incubation period at 37 °C, shaking at 200 rpm, 100 µL of culture was transferred to fresh MHII broth containing the same concentration of SAA or vehicle control or no drug. This process was repeated every 24 h over a 7-day period. On day 1 and 7, cultures were monitored for resistance development by serially diluting and spread plated in technical duplicate onto MHII-agar containing 256 µg/mL SMX or no drug. Following an 18 h incubation, presumptive SAA-selected SMX-resistant colonies were randomly selected and patched onto MHII agar plates with or without 256 µg/mL or 512 µg/mL of SMX to confirm SMX resistance. Four independent replicates were performed. A total of 20 SMX-resistant mutants (5 from each biological replicate) were screened for chromosomal *folP* mutations by amplifying the *folP* gene using colony PCR followed by gel-purification and DNA sequencing. A 50 µL colony PCR reaction mixture contained 10 µL of the chromosomal DNA solution as the template, 0.5 µM of each primer (*folP* For and *folP* Rev, Table S6), 0.2 mM dNTPs, 1x Phusion HF buffer, 5% DMSO, and 1 U of Phusion DNA polymerase (New England Biolabs). The mixture was heated for 30 s at 98 °C, followed by 35 cycles of 30 s at 95 °C, 30 s at 65 °C, 30 s at 72 °C, and concluding with 7 min at 72 °C. The *folP*-containing PCR product was gel-purified and sequenced to confirm the presence of *folP* mutations.

To assess if the identified *folP* chromosomal mutations contributed to sulfa resistance, confirmed *folP* mutants were colony PCR amplified using primers carrying a C-terminal FLAG sequence fusion so that DHPS production levels could be monitored, and subcloned into the expression vector pGDP2. Colony PCR was performed in a similar manner as described above except the following primers were used to amplify the mutated chromosomal *folP* gene: $folP_{ins188FG}$ For and 200 µM $folP_{ins188FG}$ Rev (Table S6). PCR amplicons were column purified, restriction-digested with HindIII and XbaI and ligated into HindIII-XbaI restricted, alkaline dephosphorylated pGDP2. Ligated plasmid DNA was transformed into *E. coli* DH5α and transformants were selected on 25 µg/mL kanamycin. Plasmid DNA was recovered and sequenced. The resultant plasmid, pMJF224 was introduced into $CaCl_2$-competent *E. coli* Δ*folP* via heat-shock and plasmid-bearing cells were selected on L-agar plates containing 25 µg/mL kanamycin.

## Molecular dynamics simulations

MD simulations were executed using the GROMACS package version 2020.2[86]. The system was set inside a dodecahedron box. The CHARMM forcefield was selected along with the TIP3P water model. An energy minimization step was conducted employing the Steepest Descent algorithm with up to 50000 steps. For the isothermal-isometric (NVT) ensemble, the leap-frog integrator algorithm was run with a time step of 2 fs. Bonds were constrained using the LINCS algorithm and the long-range electrostatic interactions were calculated using the Particle Mesh Ewald (PME) method. The Maxwell distribution was used to generate the initial velocity from a random seed. For the isothermal-isobaric (NPT) ensemble, the V-rescale method was used for temperature coupling and the Berendsen method for pressure coupling. The MD simulation was performed for 200 ns and the MD simulation trajectories were analyzed using functions including the gmx rmsf command.

## Reporting summary

Further information on research design is available in the Nature Portfolio Reporting Summary linked to this article.

## Data availability

The crystallographic data generated in this study have been deposited in the Protein Data Bank under accession codes 7S2I, 7S2J, 7S2K, 7S2L, 7S2M, 7TQ1 and 8SCD. Source data are provided with this paper.

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

## Acknowledgements

We thank Rosa Di Leo for assistance with cloning. We thank the staff of the Structural Biology Center, Advanced Photon Source, Argonne National Laboratory (Youngchang Kim, Kemin Tan, and especially Karolina Michalska) for X-ray diffraction data collection and/or data processing. We thank Robert Flick and the BioZone Mass Spectrometry Facility for analysis of synthesis of DHPP and evaluation of formation of SMX-pterin adduct. We thank Gerry Wright for providing plasmid pGDP2. This research was also enabled in part by support provided by SciNet (www.scinethpc.ca) and the Digital Research Alliance of Canada (www.alliancecan.ca). Computations were performed on the Graham supercomputer at the SciNet HPC Consortium. SciNet is funded by: the Canada Foundation for Innovation; the Government of Ontario; Ontario Research Fund—Research Excellence; and the University of Toronto. This work has been funded in whole or in part with U.S. Federal funds from the National Institute of Allergy and Infectious Diseases, National Institutes of Health, Department of Health and Human Services, under Contract Nos. HHSN272201700060C and 75N93022C00035 (Center for Structural Biology of Infectious Diseases, CSBID) to A.S. This project was funded in part by an AAFC project grant to M.F.

## Author contributions

M.V. performed enzyme activity assays, tryptophan fluorescence experiment, crystallography, mass spectrometry experiments, analyzed data and wrote the manuscript; M.F. generated the *E. coli* ΔfolP strain, characterized strain growth, performed antimicrobial susceptibility assays, performed the adaptive laboratory evolution experiment, analyzed data and wrote the manuscript; L.A.V. assisted with the work of M.F.; T.S. performed crystallography experiments; N.M. assisted with protein purification; R.F. performed mass spectrometry and analyzed data; C.P. performed molecular dynamics simulations; R.M. supervised C.P.; P.J.S. conceived of the project, solved crystal structures, analyzed protein structures, analyzed data, and wrote the manuscript, A.S. analyzed data and wrote the manuscript.

## Competing interests

The authors declare no competing interests.
