## [Peer Review File · Nature Communications]

Molecular mechanism of plasmid-borne resistance to sulfonamide antibioticsReviewer #1 (Remarks to the Author):

This is an important manuscript that explains the molecular mechanism of plasmid-borne resistance to sulfonamide antibiotics. Sulfonamides are very widely used in human and animal health. Their use for treating Gram-Negative infections has been severely compromised by widespread resistance, which is concerning as they are one of the few cheap, orally bioavailable agents suitable for use in the outpatient setting. The authors examine the biochemical, structural, and microbiological basis for the sulfonamide resistance caused by the plasmid-borne sul enzymes at a level of detail that has not previously been reported for studies of these enzymes. The study clearly demonstrates how the sul enzymes differentiate sulfonamides over the native substrate pABA, and further shows how Phe-Gly DHPS insertion into DHPS recapitulates this selectivity and consequent resistance.

The study is sound, with the redundancy of biochemical and microbiological studies providing high confidence in the results, reproducibility, and resulting conclusions. The manuscript is well-written and easy to follow and should become highly cited by the antimicrobial resistance community.

General comments for consideration:

1. The authors should address if there is a difference in the level of expression of the plasmid-borne Sul enzymes vs. native DHPS in resistant clinical isolates.
2. The study is written from a static structural perspective despite the resistance elements being found in highly dynamic regions of the proteins. It would be inciteful to know how the Phe-Gly insertion affects the loop dynamics by examination of the loop B factors or ideally through MD simulations of the ternary complex. How the insertions affect the overall stability of the enzymes should also be considered.
3. The sul selectivity against sulfanilamide (the smallest of the sulfonamides), which shares a similar molecular envelope to pABA is intriguing. Supplementary figure S8 should include the corresponding dihydropteroate structure and measure the carboxylate distance to F179. If similar what is the justification?

Suggested minor revisions:

1. Line 242: Mentions that loop 3 maintains its conformation in the apo vs. ligand-bound state. Then later in the paper, it is shown that loop 3 changes conformation.
2. Line 263: figure 2d, is should it be 2c.
3. Line 332: Sul2 should be Sul3
4. Line 372: Is the text referring to figure 4C,F instead of 5C,F
5. Line 523: Molecular dynamics simulations may be helpful here.
6. Line 500: The most frequently found mutation was at T62. Looking at the sequence alignment of DHPS and the different Sul enzymes in supplementary figure S7, I do not see a T62 at that site.

In conclusion, this is an excellent study, well worth publication, congratulations to the authors.

Richard E. Lee

Reviewer #2 (Remarks to the Author):

Validity

The paper Molecular mechanism of plasmid-borne resistance to sulfonamide antibiotics by Venkatesan et al., have analysed the molecular foundation for Sul-mediated drug resistance and for development of reimagined sulfas less prone to resistance. Methods applied are crystal structure complexes, biochemical and biophysical assays, mutational analysis and in trans complementation mutations in E. coli of several genes.

Significance

The authors have described the problem of sulfonamides resistance through sul genes, in pathogenic bacteria, in environmental strains to put the problem into perspective. Also the sulfamethoxazole is found in waste-water as pollution indicating a strong selective pressure on bacteria to acquire the sul genes.

The paper is sound and it the first molecular study of Sul enzymes to shed light on the mechanism by which sulfonamides resistance is conferred by the enzymes Sul1, Sul2 and/or Sul3.

Sulfonamides work by inhibiting dihydropteroate synthase (DHPS) enzyme through chemical mimicry of its co-substrate p-aminobenzoic acid.

Data and methodology and Analytical approach

The methods and data analysis applied are through the paper is adequate.

The methods go from the bacterial level down to the atomic 3D structural level and conformational dynamics. The analytical approach is appropriate.

Suggested improvements

The paper is well written with few errorer.

-The presentation of the result section must be less wordy and less detailed to bring the reader more easily through all the results.

-In the reference list all specie names must be in italic, and the look over that all gene names have the correct "layout/nomenclature".

In the reference list, all "in vivo" or in vitro" must be in italic

L573 explain how the conformational changes were observed. The current sentence "..monitor conformational changes.." is not clear.

Minor corrections:

- throughout the text, be consistent to use "space" or " not space" between degrees and the number

- throughout the text, be consistent to use "space" or " not space" between unit and number (e.g. L795 (1h), L843 (60s), L867 (50mm etc) and more of these.

- L75 remove space between reference 13,14 and period.

There are enough details in the material and methods for others to reproduce the data.

Reviewer #3 (Remarks to the Author):

Venkatesan et al provide an in-depth study on the mechanism of Sul enzymes, which catalyze dihydropteroate formation but are refractory to the action of sulfonamide antibiotics, so their production in Gram-negative results in sulfonamide resistance. Crystal structures with various ligands/substrates show important loop movements and guide further biophysics, biochemical and microbiological studies. Together, the data are convincing that a Phe-Gly insertion, and loop dynamics, contribute to the observed substrate discrimination of Sul enzymes. These data will have important implications for the design of new sulfonamide antimicrobials.

The experiments are well thought through and nicely presented, and I would recommend publication in Nature communications, after the below (mostly minor) comments have been addressed:

1. The paper is well written but is a little too long for a communications journal (~8000 words at my count), meaning it can be hard to follow at times. Some careful editing to reduce the word count would help.

2. To this end, fully understanding the mechanisms being discussed, particularly for a more general audience, would benefit from a more complete Figure 1 that presents a more detailed chemical mechanism (e.g. including intermediates that are known to form during the reaction). This would help understand all the different nomenclature that is used for the ligands, and the proposed parts of the reaction that have been captured.
3. For example, lines 277 – 284 are not very clear. What do the authors mean they have captured the SN1 reaction mechanism, and the Michaelis transition state. Perhaps a figure would help to explain which bit of the mechanism this is a snapshot of, and what the transition state is? There should probably also be a figure to show the superimposition of the Sul3 complex with the previously published YpDHPS complex.
4. In 7S2O, the ligand (XHP) has a very low RSCC (0.71), indicative of a very poor fit to the density, especially for a small ligand with no discernable flexible regions. The density presented in Figure S5d also isn't very convincing, even if it's a simulated annealing omit map. How can the authors be sure this is present and modelled correctly, and they are not modelling in to noise? Maybe a Polder map could help here? Which parts of the molecule in particular are not visible in the electron density? Could it be in more than one orientation or very mobile? If this molecule subsequently needs to react with pABA, is it odd that it's not well resolved here? I think if there isn't any other strong evidence (e.g. is there any mass spec evidence this is produced in Sul enzymes?) then it should be removed from the final structure and in the manuscript it should be referred to as unmodelled density that could be DHP+. Also, the modelled isopropanol (IPA) ligand is at 0.25 RSCC – this should be removed from the structure and then resubmitted to the PDB for a new validation report.
5. Line 220 mentions there are apo structures of Sul1, Sul2 and Sul3, but no Sul1 apo structure has been presented? Having an apo structure of Sul1 in the same space group as the liganded structure would be important to understand the associated loop movements.
6. Three structures have 6-HMP bound, but at no point is it mentioned what this is. There should be a chemical drawing of the ligand somewhere, and an explanation of what it is and why it is used (is it inhibitory, is it a known mimic of a known intermediate in the reaction?).
7. On line 242 one of the structures is called the Sul3·6-MP·Mg²⁺ 242 ·PPi; in line 270 the same structure (I think) is called Sul3·DHPP·Mg²⁺ 270 ·pABA complex; in figures the Sul3·DHP+·pABA·Mg²⁺·PP complex. The naming should be more consistent. Also, the 6-MP / DHP+ nomenclature is confusing, they seem to be labelled interchangeably; to help the reader there should be a chemical structure of it, perhaps as part of a more detailed Figure 1 if it's a reaction intermediate.
8. As the structures are all in different space groups it is worth mentioning whether the discussed loop movements are affected by crystal contacts (particularly in lines 235 to 245). In Figure S5a, for example, its shown that loops 1 and 2 are stabilised by crystal contacts, presumably loop 3 isn't involved in crystal contacts?
9. On line 277 it mentions Figure S5f, but there isn't a panel 'f' in Figure S5. I think the authors mean panel 'd'?
10. The PDB validation report for 7TQ1 is missing for me.
11. Figure S7 comes before S5, so the supplemental figures should be renumbered as they appear in the text.
12. It's not clear how the models in Figure S8 were generated. Have they been energy minimised in anyway, or undergone any molecular dynamics? If they haven't been, how have the geometries of the sulfa ligands (particular their side chains) been determined? Is there any correlation between their inhibitory effects and size/chemical identity of the side chain?

13. Refinement of 7S2L appears to be at a higher resolution (2.79) than the data were collected at (2.8), which isn't possible?

Point-by-point Response to Reviewers for “Molecular mechanism of plasmid-borne resistance to sulfonamide antibiotics”

Bold = reviewer comments

Not bold = our reply.

Reviewer #1 (Remarks to the Author):

This is an important manuscript that explains the molecular mechanism of plasmid-borne resistance to sulfonamide antibiotics. Sulfonamides are very widely used in human and animal health. Their use for treating Gram-Negative infections has been severely compromised by widespread resistance, which is concerning as they are one of the few cheap, orally bioavailable agents suitable for use in the outpatient setting. The authors examine the biochemical, structural, and microbiological basis for the sulfonamide resistance caused by the plasmid-borne *sul* enzymes at a level of detail that has not previously been reported for studies of these enzymes. The study clearly demonstrates how the *sul* enzymes differentiate sulfonamides over the native substrate pABA, and further shows how Phe-Gly DHPS insertion into DHPS recapitulates this selectivity and consequent resistance.

The study is sound, with the redundancy of biochemical and microbiological studies providing high confidence in the results, reproducibility, and resulting conclusions. The manuscript is well-written and easy to follow and should become highly cited by the antimicrobial resistance community.

General comments for consideration:

1. The authors should address if there is a difference in the level of expression of the plasmid-borne *Sul* enzymes vs. native DHPS in resistant clinical isolates.

We thank the reviewer for raising this point. We agree that expression levels of the *sul* and *folP* genes in resistant clinical isolates could provide some insight, however we believe that properly addressing this issue is outside the scope of the current manuscript, which was more focused on the *Sul* enzyme structure and function. This question would necessitate an extensive research study on its own and we do think this is an important research avenue for the future.

Even in the literature, very little is known about the expression levels as well as the genetic regulation of the combination of the chromosomally-encoded *folP* gene and plasmid-borne *sul* genes in both lab or clinical isolates of *E. coli* or other species. It is known that the *sul1* or *sul2* gene on plasmids re expressed from their own promoters (Guerineau, F. et. al. 1990. 23(1):35-41, <https://journals.asm.org/doi/epdf/10.1128/AAC.32.11.1684>) or they can be expressed from promoters that drive the expression of multiple resistance genes (Swedberg, G. 1987, Antimicrob. Agents. Chemother. 31(2):306-311, Sundstrom et. al. 1988. Mol Gen Genet. 213:191-201).

Based on data we did not include in this paper, but we may include in a future publication more focused on the expression and regulation of these genes, we believe that there must be fine-tuning of their relative expression. This is because, in our hands, over-expression of *folP*, *sul1*, *sul2* or *sul3* in *E. coli* from a high-copy plasmid (pUC19) had a negative impact on bacterial growth and actually increased susceptibility to sulfamethoxazole. In agreement with this, Zhou et. al 2021 demonstrated that overexpression of *sul3* in *E.*

coli had a negative impact on bacterial growth (<https://ami-journals.onlinelibrary.wiley.com/doi/full/10.1111/1462-2920.15783>). Since this observation would have complicated our mutational analysis of the EcDHPS and Sul enzymes, we switched to the low-copy plasmid pGDP2 for this paper and did not present the results from pUC19. As mentioned, we hope to address the expression and regulation issue in a future publication.

2. The study is written from a static structural perspective despite the resistance elements being found in highly dynamic regions of the proteins. It would be inciteful to know how the Phe-Gly insertion affects the loop dynamics by examination of the loop B factors or ideally through MD simulations of the ternary complex. How the insertions affect the overall stability of the enzymes should also be considered.

We absolutely agree with the reviewer that dynamics plays an important role in Sul-mediated resistance. Our crystal structures which showed different conformations of the alpha6 and Phe-Gly region depending on the ligand-bound state, and our Trp fluorescence experiments hinted at this. Like our answer to the question above, we believe analysis of the flexibility of this region of the Sul enzymes warrants an in-depth research study on its own using biophysical approaches as NMR or H/D exchange mass spectrometry. However, we did perform some of the analysis the reviewer suggested: analysis of B-factors and molecular dynamics.

On the B-factor side, we noticed that the B-factors for the alpha6 region (containing the key Phe residue for drug resistance) and the sidechain of the Phe residue itself were generally higher than the average of the entire protein. For example, in the Sul2 apo structure, the average B-factor for the entire protein was 36, the range of B-factors of alpha6 was between 20 and 53, and the B-factor of the sidechain of the key F179 residue was between 32 and 67. Also in the Sul2+7,8-dihydropteroate structure, the average B-factor for the entire protein as 31, the range of B-factors for alpha6 was between 17 and 48, and the B-factor of sidechain of F179 was between 26 and 48. This suggests that alpha6 and the sidechain of the key Phe residue may be more structurally dynamic than the average of the while protein.

With regards to molecular dynamics simulations, we conducted 200 ns simulations of the Sul2 WT, Sul2 F179G variant, EcDHPS WT and EcDHPSinsFG variant. We measured the RMSF (root mean square fluctuation) of the Ca atoms of the $\alpha 6$ region in each enzyme and then compared the WT vs. variants. We noticed that the F179G substitution in Sul2 decreased the RMSF of residues in $\alpha 6$ (residues 176-184), suggesting that this substitution decreases the flexibility of this area. The Ca of F179 was not significantly changed. This was also true for EcDHPS, where the MD showed that the insertion FG at position 190 decreased the RMSF of the $\alpha 6$ region in this enzyme (187-195). Our interpretation of these simulations is that the F179G supports a more organized binding site for sulfa compound engagement as compared to the WT enzyme, and for EcDHPS, the insertion of FG may allow for $\alpha 6$ to adopt a specific conformation to occlude sulfa binding.

Based on these observations we updated the Results:

Inserted the following around line 246:

In general, the sidechain of this residue and its environment ($\alpha 6$) showed higher crystallographic *B*-factors than the average protein (i.e. for the Sul2 apo structure, the *B*-factor of the Phe179 sidechain atoms were 32 to 67, for $\alpha 6$: 20 to 53, and for the protein average: 36).

And we inserted the following to the end of the Results section:

Finally, we utilized molecular dynamics simulations to gauge whether variations in the sequence composition of the $\alpha 6$ region of Sul and *EcDHPS* affect its flexibility. Simulations of the Sul2 WT, Sul2^{F179G}, *EcDHPS* and *EcDHPS*^{insFG} *apo* enzymes revealed that both this point substitution in Sul2 and the insertion in *EcDHPS* decreased $\alpha 6$'s flexibility (root-mean-square-deviation, **Supplementary Information S15**). These results suggested that Sul2^{F179G} harbors a more organized binding site for sulfa drug binding, while *EcDHPS*^{insFG} provides a capability of this enzyme to adopt a specific conformation to occlude sulfa binding.

3. The sul selectivity against sulfanilamide (the smallest of the sulfonamides), which shares a similar molecular envelope to pABA is intriguing. Supplementary figure S8 should include the corresponding dihydropteroate structure and measure the carboxylate distance to F179. If similar what is the justification?

We thank the reviewer for this astute observation, and we also agree that it is intriguing that the Sul enzymes are resistant to sulfanilamide even though it is the most similar in structure to pABA. It appears that reason for this is that the sulfonamide nitrogen itself clashes with F179. This is what was labeled on Supplementary Figure S8 with a double-ended arrow and distance of 2.1 Å and indicated in the main text. Since this amide nitrogen is present in all sulfonamides, we expect it to clash with F179 in all cases, along with clashes between the different substituent groups off the sulfonamide nitrogen with F179. We modified the main text accordingly:

“This analysis showed that all 12 compounds would pose a steric clash with the Phe residue of $\alpha 6$ helix, including the most primitive sulfa sulfanilamide, whose structure is the closest mimic to pABA. The modeling suggested that the sulfonamide nitrogen would be positioned 2.1 Å distance from the Phe residue, resulting in an unfavourable clash. The modeling further suggested that the N-acyl substituents off the sulfonamide nitrogen that define the different compounds would exacerbate the clash with F179, at least using their energy-minimized conformations in our models.”

We also updated Supplementary Figure S8 to show the carboxylate distance between pABA and F179.

Suggested minor revisions:

1. Line 242: Mentions that loop 3 maintains its conformation in the apo vs. ligand-bound state. Then later in the paper, it is shown that loop 3 changes conformation.

We agree, this was incorrect and this sentence was removed:

In contrast, in the Sul2·7,8-DHP·Mg²⁺·PP_i complex and Sul3·DHP⁺·pABA·Mg²⁺·PP_i complexes, loops 1 through 3 retain the same conformation as in the corresponding *apo* structures.

2. Line 263: figure 2d, is should it be 2c.

This was corrected to 2c.

3. Line 332: Sul2 should be Sul3

Corrected.

4. Line 372: Is the text referring to figure 4C,F instead of 5C,F

Corrected to Figure 4C, 4F.

5. Line 523: Molecular dynamics simulations may be helpful here.

Please see our reply to comment #2 above where we explain how we simulated the dynamics of the *EcDHPSinsFG* structure.

6. Line 500: The most frequently found mutation was at T62. Looking at the sequence alignment of DHPS and the different Sul enzymes in supplementary figure S7, I do not see a T62 at that site.

Apologies that this was unclear, it may have been due to the residue numbering on S7 which was based on Sul1. We added a green box around position 72 in *EcDHPS*, updated its figure legend, and updated the main text to read:

Among all the mutations found in the coding region of *EcDHPS*, the most frequently observed mutations involved Thr62 (located in loop 2, 15/20 isolates) (**Figure 5A, Supplementary Information S5**).

In conclusion, this is an excellent study, well worth publication, congratulations to the authors.

Richard E. Lee

Richard, thank you very much for the kind words.

Reviewer #2 (Remarks to the Author):

Validity

The paper Molecular mechanism of plasmid-borne resistance to sulfonamide antibiotics by Venkatesan et al., have analysed the molecular foundation for Sul-mediated drug resistance and for development of reimagined sulfas less prone to resistance. Methods applied are crystal structure complexes, biochemical and biophysical assays, mutational analysis and in trans complementation mutations in *E. coli* of several genes.

Significance

The authors have described the problem of sulfonamides resistance through sul genes, in pathogenic bacteria, in environmental strains to put the problem into perspective. Also the sulfamethoxazole is found in waste-water as pollution indicating a strong selective pressure on bacteria to acquire the sul genes.

The paper is sound and it the first molecular study of Sul enzymes to shed light on the mechanism by which sulfonamides resistance is conferred by the enzymes Sul1, Sul2 and/or Sul3. Sulfonamides work by inhibiting dihydropteroate synthase (DHPS) enzyme through chemical mimicry of its co-substrate p-aminobenzoic acid.

Data and methodology and Analytical approach

The methods and data analysis applied are through the paper is adequate.

The methods go from the bacterial level down to the atomic 3D structural level and conformational dynamics. The analytical approach is appropriate.

Suggested improvements

The paper is well written with few errorer.

-The presentation of the result section must be less wordy and less detailed to bring the reader more easily through all the results.

We agree with this comment and the paper has been shortened from 8128 words to 6630 words.

-In the reference list all specie names must be in italic, and the look over that all gene names have the correct "layout/nomenclature".

In the reference list, all "in vivo" or in vitro" must be in italic

We corrected species names, gene names and "in vivo"/"in vitro" throughout the paper to be italicized.

L573 explain how the conformational changes were observed. The current sentence "...monitor conformational changes.." is not clear.

We elaborated on monitoring of conformational changes in the enzyme using ITF and changed line 572-574 to read:

By introducing a single tryptophan residue at the appropriate structural elements of Sul and *Ec*DHPS enzymes we were able to identify that conformational changes occur around the active site of these enzymes during catalysis via changes in the intrinsic tryptophan fluorescence signal

Minor corrections:

- throughout the text, be consistent to use “space” or “ not space” between degrees and the number

This has been corrected to be “space” between degree symbol and number.

- throughout the text, be consistent to use “space” or “ not space” between unit and number (e.g. L795 (1h), L843 (60s), L867 (50mm etc) and more of these.

This has been corrected to be “space” between unit and number.

- L75 remove space between reference 13,14 and period.

This has been corrected.

There are enough details in the material and methods for others to reproduce the data.

Reviewer #3 (Remarks to the Author):

Venkatesan et al provide an in-depth study on the mechanism of Sul enzymes, which catalyze dihydropteroate formation but are refractory to the action of sulfonamide antibiotics, so their production in Gram-negative results in sulfonamide resistance. Crystal structures with various ligands/substrates show important loop movements and guide further biophysics, biochemical and microbiological studies. Together, the data are convincing that a Phe-Gly insertion, and loop dynamics, contribute to the observed substrate discrimination of Sul enzymes. These data will have important implications for the design of new sulfonamide antimicrobials.

The experiments are well thought through and nicely presented, and I would recommend publication in Nature communications, after the below (mostly minor) comments have been addressed:

1. The paper is well written but is a little too long for a communications journal (~8000 words at my count), meaning it can be hard to follow at times. Some careful editing to reduce the word count would help.

We agree with this comment and the paper has been shortened from 8128 words to 6630 words.

2. To this end, fully understanding the mechanisms being discussed, particularly for a more general audience, would benefit from a more complete Figure 1 that presents a more detailed chemical mechanism (e.g. including intermediates that are known to form during the reaction). This would help understand all the different nomenclature that is used for the ligands, and the proposed parts of the reaction that have been captured.

We appreciate this suggestion and have updated Figure 1 to show the expected reaction mechanism, including the intermediate DHP⁺, and also to show the structure of 6-HMP to address comment #6 from this reviewer. The figure legend has been updated accordingly.

3. For example, lines 277 – 284 are not very clear. What do the authors mean they have captured the SN1 reaction mechanism, and the Michaelis transition state. Perhaps a figure would help to explain which bit of the mechanism this is a snapshot of, and what the transition state is? There should probably also be a figure to show the superimposition of the Sul3 complex with the previously published YpDHPS complex.

After addressing the reviewer's point #4 below and identifying a crystal with much stronger and obvious electron density in the active site of the Sul3 complex, it is now clear to us that we have not trapped a transition state but instead an intermediate in the SN1 reaction mechanism, where the pyrophosphate of the co-substrate DHPP has been lost, leaving the compound DHP⁺. We updated Figure 1 to illustrate this reaction mechanism and the chemical structure of DHP⁺ more clearly. We also updated lines 270-280 to more clearly explain this:

The Sul3·DHP⁺·Mg²⁺·*p*ABA complex structure also featured a well-defined active site. This Sul complex structure contained electron density corresponding to DHP⁺²¹, *p*ABA, Mg²⁺ and PP_i (**Supplementary Information Figure S6d**). Multiple electron density features verify this trapped molecule is DHP⁺, including a clear break between its C9 position and the *p*-amino group of *p*ABA, and a ring pucker around atoms C6, C7 and N8 consistent with saturation of at the C7 position (**Supplementary Information Figure S6d**). These observations are consistent with *in*

crystallo-capture of the intermediate of the S_N1 enzymatic reaction mechanism, wherein the PP_i group has been cleaved from DHPP but the *p*ABA molecule had not yet been ligated to DHP⁺.

Note that Supplementary S5 is now S6 due to reordering.

Since we switched to referring to DHP⁺ as an intermediate, the double dagger (‡) label was removed from Figures 2, 3, 4 and Supplementary Figures.

We also modified Figure 3 to show a superposition between the Sul3·DHP⁺·Mg²⁺·*p*ABA complex and the published YpDHPS·DHP⁺·Mg²⁺·*p*ABA complex. To balance the positioning of the panels in Figure 3, panel 3D was moved to the left. The main text was modified to reference the new panel Figure 3F:

The position of DHP⁺ in the Sul3·DHP⁺·*p*ABA·Mg²⁺·PP_i complex superimposed with its position in the YpDHPS transition state complex structure (**Figure 3f**)²¹ and the pterin ring in the Sul2·7,8-DHP·Mg²⁺·PP_i complex.

4. In 7S2O, the ligand (XHP) has a very low RSCC (0.71), indicative of a very poor fit to the density, especially for a small ligand with no discernable flexible regions. The density presented in Figure S5d also isn't very convincing, even if it's a simulated annealing omit map. How can the authors be sure this is present and modelled correctly, and they are not modelling in to noise? Maybe a Polder map could help here? Which parts of the molecule in particular are not visible in the electron density? Could it be in more than one orientation or very mobile? If this molecule subsequently needs to react with *p*ABA, is it odd that it's not well resolved here? I think if there isn't any other strong evidence (e.g. is there any mass spec evidence this is produced in Sul enzymes?) then it should be removed from the final structure and in the manuscript it should be referred to as unmodelled density that could be DHP⁺. Also, the modelled isopropanol (IPA) ligand is at 0.25 RSCC – this should be removed from the structure and then resubmitted to the PDB for a new validation report.

We agree with the reviewer that the electron density for the DHP⁺ compound (PDB compound code XHP) was weak. At the time we had collected x-ray data for multiple crystals and this one was chosen for the final structure, but in retrospect this was done in haste. We analyzed the data from all our other crystals and discovered one that had much better electron density for the DHP⁺ compound. It was actually in the same conformation as the one we initially modeled for 7S2O, and the rest of the structure is unchanged, but the density with this other crystal is much more convincing. We retracted the 7S2O structure from the PDB and deposited this new structure under the accession code 8SCD, and the validation report is attached – the RSCC for DHP⁺ (PDB chemical code XHP) is much improved at 0.89 and the ligand refined to a better occupancy of 0.80 (as compared to 0.65) We updated all figures with this structure accordingly (Figure 2, Figure 3A, Figure 3C, Figure 3F, Figure 4D, Supplementary Information Figure S6, Supplementary Information Figure S7, Supplementary Information Figure S9C) and the materials section with the occupancy value of 0.80.

5. Line 220 mentions there are apo structures of Sul1, Sul2 and Sul3, but no Sul1 apo structure has been presented? Having an apo structure of Sul1 in the same space group as the liganded structure would be important to understand the associated loop movements.

The sentence at lines 219-222 was poorly worded as it suggested that we obtained a crystal structure of Sul1 in the apo state, which we did not. The wording for this sentence was adjusted to the following:

We determined the crystal structure of Sul1 in a ligand bound state, and the Sul2 and Sul3 enzymes in both *apo* and ligand bound forms to resolutions between 1.74 to 2.8 Å (**Table S1**).

We agree that the proper comparison for understanding loop movements is between the apo and various ligand bound states. These were available for Sul2 and Sul3 and so our discussion of conformational changes in lines 235-253 was focused on these two enzymes; we assume these are representative of Sul1 also (since Sul1 is 54% identical, 69% similar in sequence to Sul2) but we kept our discussion to Sul2 and Sul3 since those actually yielded apo crystal structures. No changes to the text were made in lines 235-253.

6. Three structures have 6-HMP bound, but at no point is it mentioned what this is. There should be a chemical drawing of the ligand somewhere, and an explanation of what it is and why it is used (is it inhibitory, is it a known mimic of a known intermediate in the reaction?).

We apologize for not explaining this compound fully. 6-HMP is 6-hydroxymethyl-7,8-dihydropterin. We incubated Sul3 and EcDHPS^{insFG} variant with 6-hydroxymethyl-7,8-dihydropterin pyrophosphate (DHPP). Upon solving the structures, density only for the pterin ring was observed and we modeled these as 6-HMP; the pyrophosphate must have been cleaved off by the enzyme.

To better explain the identity of this compound, we updated Figure 1 to show its structure and acronym. To better explain how it was observed in the structures with Sul1 and EcDHPS^{insFG}, we changed lines 254-256 to:

We identified electron density in the active site of Sul1 corresponding only to a pterin ring which we modeled as 6-hydroxymethylpterin (6-HMP, **Figure 1, Supplementary Information Figure S6a**).

(Note that Supplementary S5 is now S6 due to reordering.)

And we changed lines 510-512 to:

The crystal contained electron density in the active site corresponding only to the pterin ring which we modeled as 6-HMP (**Fig. 1, Fig. 5C, Supplementary Information Figure S6e**).

And updated the Crystallization section of the Methods to add:

Electron density features in the active sites of Sul1 and EcDHPS^{insFG} crystals was modeled as 6-HMP, likely resulting from the loss of pyrophosphate from the DHPP in the crystallization mixture.

7. On line 242 one of the structures is called the Sul3·6-MP·Mg²⁺·242·PPi; in line 270 the same structure (I think) is called Sul3·DHPP·Mg²⁺·270·pABA complex; in figures the Sul3·DHP+·pABA·Mg²⁺·PP complex. The naming should be more consistent. Also, the 6-MP / DHP+ nomenclature is confusing, they seem to be labelled interchangeably; to help the reader there should be a chemical structure of it, perhaps as part of a more detailed Figure 1 if it's a reaction intermediate.

We apologize for this confusion. We removed all references to 6-MP and instead used DHP+ (i.e. lines 242, 283, 358, 811, Figure 3, Supplemental Figures S6, S7 and S9, Table S1. Figure 1 has also been modified to show the chemical structure of DHP+ and the main text has been expanded to explain that it is a reaction intermediate (see response to point #3 above).

8. As the structures are all in different space groups it is worth mentioning whether the discussed loop movements are affected by crystal contacts (particularly in lines 235 to 245). In Figure S5a, for example, its shown that loops 1 and 2 are stabilised by crystal contacts, presumably loop 3 isn't involved in crystal contacts?

We did an analysis using the PDBePISA server for evaluating participating of the loops 1 through 3 in crystal contacts. While certainly the loops participated in some crystal contacts, they were largely free to adopt different conformations, or, there were multiple copies of the protein in the asymmetric unit such that we could observe different conformations of the loops. To expand on this, and also to discuss what the reviewer is referring to regarding loop 1 and loop 2, we adjusted lines 235-245 to:

While each crystal adopted different space groups, the loops largely did not participate in crystal contacts or block active sites, except for the Sul1 *apo* structure where loops 1 and 2 interdigitated into the active site of other chains in the crystal lattice (**Supplementary Information Figure S6a**).

9. On line 277 it mentions Figure S5f, but there isn't a panel 'f' in Figure S5. I think the authors mean panel 'd'?

This was a typo and has been fixed.

10. The PDB validation report for 7TQ1 is missing for me.

We apologize that this was missed in the submission and we have uploaded this validation report in the resubmission.

11. Figure S7 comes before S5, so the supplemental figures should be renumbered as they appear in the text.

Figure S5 through S7 have been reordered.

12. It's not clear how the models in Figure S8 were generated. Have they been energy minimised in anyway, or undergone any molecular dynamics? If they haven't been, how have the geometries of the sulfa ligands (particular their side chains) been determined? Is there any correlation between their inhibitory effects and size/chemical identity of the side chain?

The chemical structures of the sulfonamides were retrieved from the Pubchem database (3D conformer, already energy minimized; we verified they were already energy minimized using Chimera also). Then each of their 4-aminobenzene group was superimposed onto the pABA region of 7,8-dihydropteroate from the Sul2 product bound structure. We did not perform any additional energy minimization or molecular dynamics of the Sul2+sulfonamide structures as we thought this would be nonsensical, since our kinetics data unambiguously showed the sulfonamides are poor substrates of the Sul enzymes (we

assume the kinetics behaviour vs Sul1 is representative of Sul2 since they are 54% identical, 69% similar in sequence). However we recognize this methodology was not explained in the paper and so we modified the caption of Figure S8 to:

Models of the sulfa compounds (shown in cyan sticks) in the active site of Sul2. The 3D conformers (already energy minimized) of the 12 sulfonamide drugs were retrieved from Pubchem. Each of their 4-aminobenzene groups were superimposed onto the pABA region of 7,8-dihydropteroate from the Sul2·7,8-DHP·Mg²⁺·PP_i complex structure. 7,8-DHP and PP_i from the Sul2·7,8-DHP·Mg²⁺·PP_i complex are shown in thin lines, except for the sulfanilamide panel where 7,8-DHP is shown in thicker sticks. Mg²⁺ shown as a black sphere. F179 and α8 are labeled. The distance between the sulfonamide nitrogen (present in all sulfas) and the closest atom of F179 is indicated with a blue double-arrow and blue text (2.1 Å) on the sulfanilamide panel. The distance between the carboxylate oxygen of 7,8-DHP and the closest atom of F179 is indicated with a black double-arrow and black text (3.2 Å) on the sulfanilamide panel.

Regarding the inhibitory effects and the size/chemical identity of the sidechain, we thank the reviewer for bringing this up. We noticed that the MICs that we measured for the sulfonamides vs. either WT *E. coli* or *E. coli* Δ*folP*/pGDP2::*sul* were positively correlated with the pKa₂ value that describes protonation/deprotonation of the sulfonamide group (see <https://www.sciencedirect.com/science/article/pii/S0045653507009368>). This was previously observed by others and modeled as related to the uptake of the anionic form of the compounds into bacterial cells (<https://www.sciencedirect.com/science/article/pii/S0045653507009368>). There is no apparent correlation between the MIC of WT *E. coli* or *E. coli* Δ*folP*/pGDP2::*sul* with the nature or size of the sidechain of the drugs, however. According to our interpretation of the reviewer's question, we believe this is what they were referring to by asking about "inhibitory effects". We decided to update the paper to mention this correlation with the pKa₂ by modifying the region around line 433:

The results demonstrate that expression of the individual *sul* genes in *E. coli* results in high levels of resistance for all 12 sulfonamide antibiotics tested. We observed a positive correlation between the MIC of the sulfas vs. WT *E. coli* and *E. coli* Δ*folP*/pGDP2::*sul* with the pKa₂ of the sulfonamide group⁶⁶; this observation has been made before⁶⁷ and correlated with accumulation of the anionic form of the sulfas into bacterial cells⁶⁶. In all instances, the MICs we measured surpassed the CLSI resistant MIC breakpoints, indicating that the *sul* genes alone are sufficient to compromise sulfa therapy⁶⁸.

13. Refinement of 7S2L appears to be at a higher resolution (2.79) than the data were collected at (2.8), which isn't possible?

This was a typo and it was corrected in Table S1, the resolution of the data collected was also 2.79.

Reviewer #1 (Remarks to the Author):

The authors have suitably addressed all prior concerns of this reviewer.

Reviewer #3 (Remarks to the Author):

The authors have addressed my concerns/comments. I'm happy for the paper to now be published.